# A toxin-antidote system contributes to interspecific reproductive isolation in rice

Shimin You [1,4], Zhigang Zhao [1,4], Xiaowen Yu [1,4], Shanshan Zhu[2], Jian Wang[2], Dekun Lei[1], Jiawu Zhou [3], Jing Li[3], Haiyuan Chen[1], Yanjia Xiao[1], Weiwei Chen[2], Qiming Wang[1], Jiayu Lu[1], Keyi Chen[1], Chunlei Zhou[1], Xin Zhang[2], Zhijun Cheng[2], Xiuping Guo[2], Yulong Ren[2], Xiaoming Zheng[2], Shijia Liu[1], Xi Liu[1], Yunlu Tian[1], Ling Jiang[1], Dayun Tao [3] ✉, Chuanyin Wu [2] ✉ & Jianmin Wan [1,2] ✉

Breakdown of reproductive isolation facilitates flow of useful trait genes into crop plants from their wild relatives. Hybrid sterility, a major form of reproductive isolation exists between cultivated rice (*Oryza sativa*) and wild rice (*O. meridionalis*, *Mer*). Here, we report the cloning of *qHMS1*, a quantitative trait locus controlling hybrid male sterility between these two species. Like *qHMS7*, another locus we cloned previously, *qHMS1* encodes a toxin-antidote system, but differs in the encoded proteins, their evolutionary origin, and action time point during pollen development. In plants heterozygous at *qHMS1*, ~ 50% of pollens carrying *qHMS1*-D (an allele from cultivated rice) are selectively killed. In plants heterozygous at both *qHMS1* and *qHMS7*, ~ 75% pollens without co-presence of *qHMS1*-*Mer* and *qHMS7*-D are selectively killed, indicating that the antidotes function in a toxin-dependent manner. Our results indicate that different toxin-antidote systems provide stacked reproductive isolation for maintaining species identity and shed light on breakdown of hybrid male sterility.

Reproductive isolation is a common phenomenon in nature and exists in different forms. Depending on species, reproductive barriers can occur either before (pre-zygotic) or after (post-zygotic) fertilization. In plants, the pre-zygotic barriers include geographical isolation, differentiations in flower habit, failure in cell-to-cell recognition between the pollen and stigma, inability in pollen germination and pollen tube growth, and defects in cell fusion between sexual gametes. For example, the speciation locus *YUP* in mokeyflowers[1], the unilaterally incompatibility system found in *Brassicaceae*[2,3], *Zea mays*[4–7] and tomato[8–10] is the well-studied pre-zygotic mechanism. Post-zygotic reproductive isolation (also called hybrid incompatibility) has been extensively studied in animals, plants and microorganisms that include hybrid sterility (HS) and

hybrid necrosis or weakness. HS is a common hybrid incompatibility. The toxin-antidote system (sometimes called killer-protector, meiotic drive, gene drive and segregation distortion in different literatures) is one of the genetic mechanisms conferring HS in animals and plants[11]. Genetic elements belonging to this group include the *segregation distorter* (*SD*) system in Drosophila[12,13], the *wtf* loci in yeast[14,15], the *peel-1/zeel-1* and *sup35/pha-1* genes in nematodes[16–18], the *t*-haplotype in mice[19–21], the *Spok*[22] and *het-s* genes[23] in *Podospora anserina*, the *Sk* genes in *Neurospora*[24], and *APOK3* in Arabidopsis[25]. In those systems the cells/gametes that do not contain a functional antidote are selectively killed. Thus, the toxin-antidote systems evolved during speciation likely play a key role to maintain species identity.

[1]State Key Laboratory for Crop Genetics & Germplasm Enhancement and Utilization, Nanjing Agricultural University, Zhongshan Biological Breeding Laboratory, Nanjing 210095, China. [2]State Key Laboratory of Crop Gene Resources and Breeding, Institute of Crop Science, Chinese Academy of Agricultural Sciences (CAAS), Beijing 100081, China. [3]Yunnan Seed Laboratory/Yunnan Key Laboratory for Rice Genetic Improvement, Food Crops Research Institute, Yunnan Academy of Agricultural Sciences (YAAS), Kunming 650200, P. R. China. [4]These authors contributed equally: Shimin You, Zhigang Zhao, Xiaowen Yu. ✉e-mail: taody12@aliyun.com; wuchuanyin@caas.cn; wanjianmin@caas.cn

Although also being common in plants, only one HS case in Arabidopsis and a few in rice have been deciphered to causal genes. The Arabidopsis *APOK3* encodes an antidote, functioning according to the toxin-antidote model to destroy the pollen-killer activity conferred by three genetically linked elements[25]. In rice, the *S27/S28*[26], *DPL1/DPL2*[27] and *DGS1/DGS2*[28] are pairs of duplicated genes, and pollens carrying malfunctioned alleles at both of the paired loci are defective in development, leading to HS in hybrid plants. The genes encoding those systems are usually located in separated loci and how they evolved during speciation has been hypothesized in the Bateson-Dobzhansky-Muller model[29,30]. Sometimes, the toxin-antidote (or killer-protector) genes are tightly linked at a single locus, including the male HS loci *qHMS7*[31], *Sa*[32], the female HS locus *S5*[33] and the female-male HS locus *S1*[34–36]. In heterozygotes, those gametes carrying non-functional antidote or protector genes are unable to develop to maturity. In a special case, the two alleles of the *Sc* locus encode homologs of DUF1618 domain protein that is essential for pollen development but are different in copy number and promoter region[37]. In F₁ hybrids (*Sc-j/Sc-i*), *Sc-i* (more copies) is expressed whereas *Sc-j* (single copy) is suppressed, such that the pollens carrying *Sc-i* are selectively transmitted. How those genes at a single locus have been co-evolved remains elusive. Of those cloned rice systems, only *qHMS7* was based on HS between wild and cultivated rice, and all the others on HS between subspecies of *O. sativa* or between the two cultivated species *O. sativa* and *O. glaberrima* (the *S1* locus). Therefore, more efforts need to be put into uncovering genetic mechanisms underlying interspecific HS between wild and cultivated rice.

There are eight *Oryza* species in the AA genome group, including the two cultivated rice species and *O. meridionalis* (*Mer*) from Australia that is considered the earliest divergent lineage around 2.93 mya[38]. We assume that cloning and characterization of reproductive isolation genes between cultivated rice and *Mer* would provide not only their evolutionary trajectory but also strategic designs to break reproductive isolation and bridge gene flow from other wild relatives downstream of *Mer* to cultivated rice. To this end, we constructed mapping populations derived from the cross between *Mer* and Dianjingyou 1 (DJY1), a *japonica* subspecies cultivar of *O. sativa*. The F₁ hybrid plants of DJY1 and *Mer* are completely male-sterile and four underlying major quantitative hybrid male sterility loci (*qHMS1*, *qHMS2*, *qHMS7* and *qHMS9*, located on chromosome 1, 2, 7, and 9, respectively) were detected[31]. Among those, *qHMS7* has been cloned and shown to encode a toxin-antidote element and kill pollens carrying the *Mer* allele[31]. The genetic mechanisms of other three loci are still unknown.

In this study, we construct a near isogenic line NIL-*qHMS1* in DJY1 background but contains the *qHMS1* locus from *Mer*, and clone *qHMS1*. The *Mer* allele of *qHMS1* encodes a functional toxin-antidote system and kills pollens carrying the non-functional DJY1 allele. The breakdown of the *qHMS1* locus is a recent event as it is only found in Asian cultivated rice species and their closest wild progenitors. When *qHMS1* is stacked with *qHMS7* in the DJY1 background, majority of pollens (~75%) in heterozygotes are aborted. This reconstructed HS mimics reproductive isolation and provides insights into how the stacked HS genetic systems contribute to maintaining species identity.

## Results

### *Mer* allele of *qHMS1* causes development defect in pollens carrying DJY1 allele

To clone *qHMS1*, we developed a near-isogenic line (NIL) harboring the *Mer* allele of *qHMS1* in the DJY1 background (designated NIL-*qHMS1*) (Supplementary Fig. 1). NIL-*qHMS1* was morphologically similar to DJY1 and produced fertile pollens, but their hybrid F₁ plants (DJY1/NIL-*qHMS1*) produced ~50% aborted pollens (Fig. 1a–d, Supplementary Fig. 2). Examination of a BC₅F₂ population revealed a clear-cut bimodal distribution in pollen fertility (Supplementary Fig. 3). Further genotyping analysis of this population with molecular markers revealed that

individuals homozygous for the *Mer* allele and those heterozygotes carrying both *Mer* and DJY1 alleles segregated at a 1:1 ratio, with very few plants (1.1%) homozygous for the DJY1 allele, suggesting that the *Mer* allele was selectively transmitted to progeny (Fig. 1e). Moreover, a reciprocal cross test between DJY1 and F₁ (DJY1/NIL-*qHMS1*) resulted in only heterozygous progeny when DJY1 was used as the female parent and ~50% DJY1/DJY1 progeny when F₁ was used as the female (Fig. 1f). In addition, the average seed setting rate of the BC₅F₂ population was >85%, indicating normal spikelet fertility regardless of genotypes (Supplementary Fig. 4). These results collectively demonstrate that the *qHMS1* locus from *Mer* encodes a selfish genetic element and controls hybrid sterility by eliminating male gametes carrying the counterpart from cultivated rice.

Histochemical investigation revealed that there was no developmental defect seen from meiosis until the early uninucleate stage in DJY1/NIL-*qHMS1* F₁ anthers (Supplementary Fig. 5). After that, however, diffused nucleus appeared in part of microspores that did not undergo first mitosis and finally developed to hollow pollens lacking nuclei and without starch accumulation (Supplementary Figs. 6, 7), although the surface and wall structure of the aborted pollens were similar to normal pollens (Supplementary Fig. 8). Thus, the pollen abortion is likely attributed to the failure of first mitosis during the gametophytic development.

### Cloning and genomic structure of *qHMS1*

To clone *qHMS1*, we first used 356 BC₅F₂ plants and mapped *qHMS1* to a region flanked by the molecular makers CH1-11 and RM3475 on chromosome 1. Using 11,800 BC₅F₃ progeny derived from BC₅F₂ plants heterozygous at *qHMS1*, we further narrowed down the locus to a 43-kb region, which was subsequently confirmed by examination of the recombinant offspring (Fig. 1g). Based on the annotation of the *O. sativa* Japonica Group IRGSP-1.0 (http://www.gramene.org/), this region is predicted to contain five open reading frames (*ORFs*) (Fig. 1h, Supplementary Table 1). Sequence comparison between *Mer* and DJY1 revealed no difference in *ORF1*, a single base change in *ORF2*, and a 615-bp deletion and five SNPs in *ORF4* (Supplementary Fig. 9a). In DJY1 *ORF3*, a single base change from G to T was found, leading to an early stop codon (Fig. 1h). In DJY1 *ORF5*, a T-to-C base substitution results in an amino acid change of Asn to Thr, in addition to a 1391-bp transposon inserted 11-bp upstream of the ATG start codon (Fig. 1h). Then RNA sequencing (RNA-seq) experiment was performed to analyze the expression of genes in the mapping region and the results showed that *ORF1-ORF3* and *ORF5*, but not *ORF4*, are expressed in developing anther (Supplementary Fig. 9b).

### *ORF3/HPT* encodes a toxin protein

To further dissect the *qHMS1* locus, we performed targeted mutagenesis to all the five genes using the CRISPR-Cas9 technology. Loss-of-function mutation of *ORF1*, *ORF2* and *ORF4* in DJY1/NIL-*qHMS1* F₁ plants did not change the semi-sterile phenotype, thus excluding their role in male sterility control (Supplementary Fig. 10). Then we focused on *ORF3* first and produced *ORF3* knock-out lines in the DJY1/NIL-*qHMS1* F₁ background using the CRISPR-Cas9 technology. We investigated three lines each representing a unique editing type in T₀ or T₁ plants and found that loss-of-function mutation of *ORF3* rescued pollen fertility (Fig. 2a–f). As expected, the segregation of three genotypes at *qHMS1* fits into the 1:2:1 ratio in the self-bred offspring of the knockout lines and the DJY1 allele of *qHMS1* was transmitted to progeny (Fig. 2g, Supplementary Table 2). Moreover, introduction of a genomic fragment harboring *ORF3* from *Mer* into DJY1 resulted in full pollen sterility, whereas knocking-out of *ORF3* in DJY1 (DJY1*orf3*) or NIL-*qHMS1* (NIL-*qHMS1orf3*) showed no impact on pollen fertility as well as plant growth and development, indicating its exclusive role as a toxin (Supplementary Fig. 11). Further, F₁ plants from the cross of DJY1*orf3* with NIL-*qHMS1* were still semi-sterile whereas those F₁ plants from the cross of

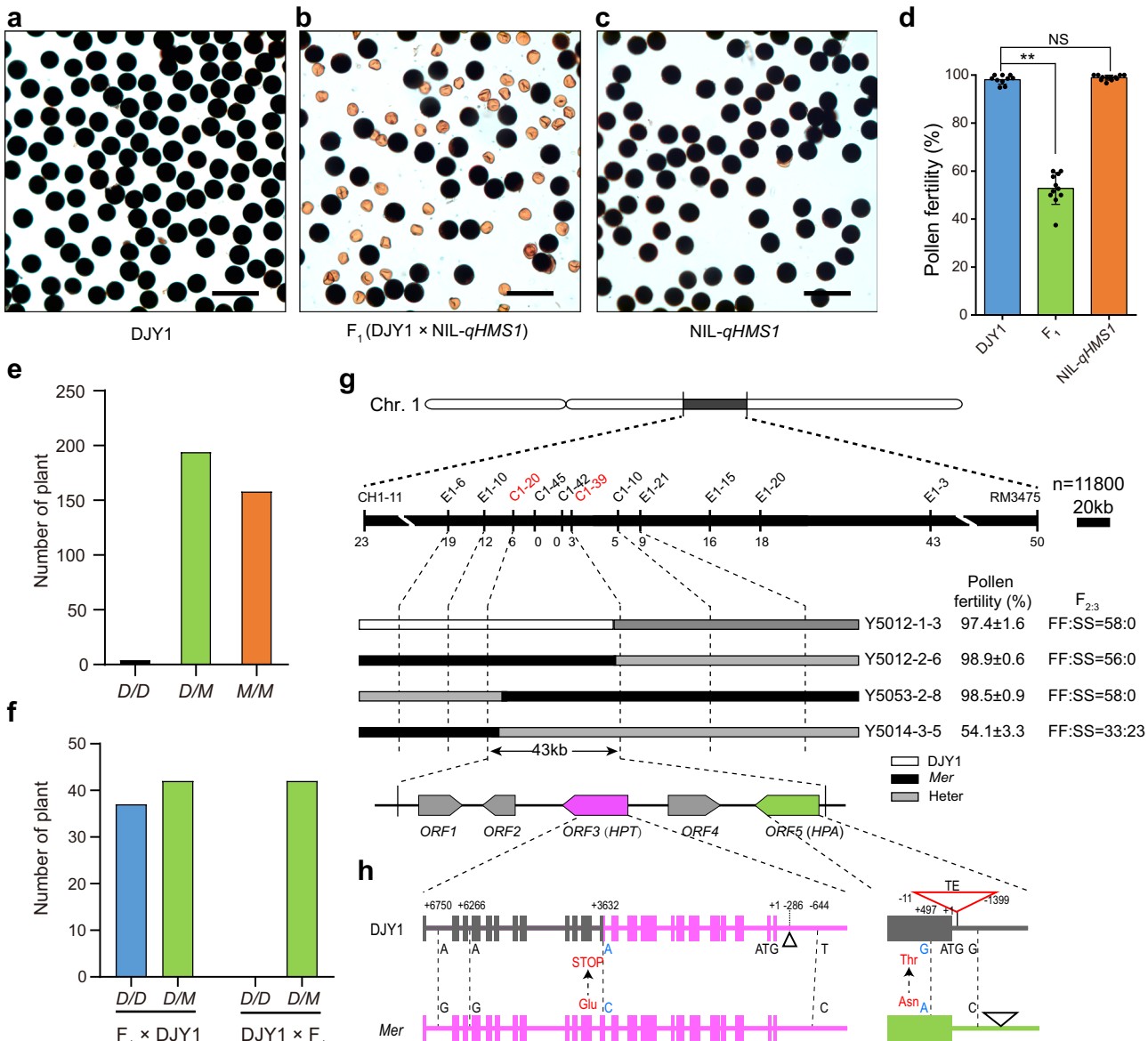

**Fig. 1 | Cloning of the *qHMS1* locus underlying the semi-sterile pollen phenotype. a–c** Normal fertile pollens in DJY1 (**a**) and NIL-*qHMS1* (**c**) and aborted pollens in their F₁ plant (**b**), as shown by I₂-KI staining to indicate starch accumulation. Scale bar, 100 µm. **d** Quantification of pollen fertility in DJY1, NIL-*qHMS1* and their F₁ shown as means ± SD (*n* = 10 plants). The semi-sterile pollen phenotype was seen in F₁. **\*\****P* = 3.47752E-9 < 0.01 by two-tailed student's *t* test; NS, no significance. **e** Genotypic distribution at *qHMS1* among a BC₅F₂ population. Few plants were found homozygous for the *D* allele, verifying the failure of its transmission through pollen. *D*, DJY1 allele; *M*, *Mer* allele. **f** Selective transmission of the *Mer* allele through pollen as tested in reciprocal crosses between DJY1 and F₁ (DJY1/NIL-*qHMS1*). Plants derived from the crosses were genotyped using molecular markers to distinguish origin of the *qHMS1* alleles. **g** *qHMS1* was anchored to a 43-kb region on chromosome 1. Numbers under the black bar indicate recombinants between markers. FF, fully fertile; SS, semi-sterile; F₂:₃, progeny of selected F₂ recombinants. Pollen fertility of these recombinants shown as means ± SD (*n* = 3 independent spikelets). **h** Genomic structure of the two candidate genes constituting a selfish genetic element. Vertical bars indicate exons. Position of single nucleotide polymorphisms (SNP) and transposon insertion is shown relative to the start codon. The two SNPs in the coding regions are highlighted in blue. TE, transposable element. Two black triangles indicate a 19-bp and a 235-bp insertion, respectively. Source data are provided as a Source Data file.

DJY1 with NIL-*qHMS1^orf3* were fully fertile, confirming that it is the *ORF3* from *Mer* that encodes a functional toxin (Fig. 2h). We thus named the gene as *HPT* (*Hybrid Pollen Toxin*) hereafter.

### *ORF5/HPA* encodes an antidote protein

The fact that HPT is not toxic to its host pollen infers that the *Mer* allele of *qHMS1* may harbor an antidote element. Accordingly, we focused on the remaining gene *ORF5* contained in the mapped region. We also used the CRISPR-Cas9 technology to produce loss-of-function mutation of *ORF5* in the same DJY1/NIL-*qHMS1* F₁ background and found that the knock-out line showed complete sterility, instead of ~50% male

sterility (Fig. 3b, e). Subsequently, we knocked out *ORF5* in both DJY1 (homozygous for the *ORF5*-D allele and fully fertile) and NIL-*qHMS1* (homozygous for the *ORF5*-*Mer* allele and fully fertile) and found that the DJY1^orf5 plant was still fully fertile but the NIL-*qHMS1^orf5* plant lost pollen fertility completely (Fig. 3a–d). Those results indicate that *ORF5* encodes an antidote detoxifying the toxin protein HPT of the *Mer* origin and that only the one from *Mer* is functional. We thus named *ORF5* as *HPA* (*Hybrid Pollen Antidote*) hereafter. This notion was supported by introduction of a *Mer* genetic fragment containing intact *HPA* into DJY1/NIL-*qHMS1* F₁, which restored male fertility from ~50% to ~75% in single-copy transgenic plants (Fig. 3f–h). Genotyping of the

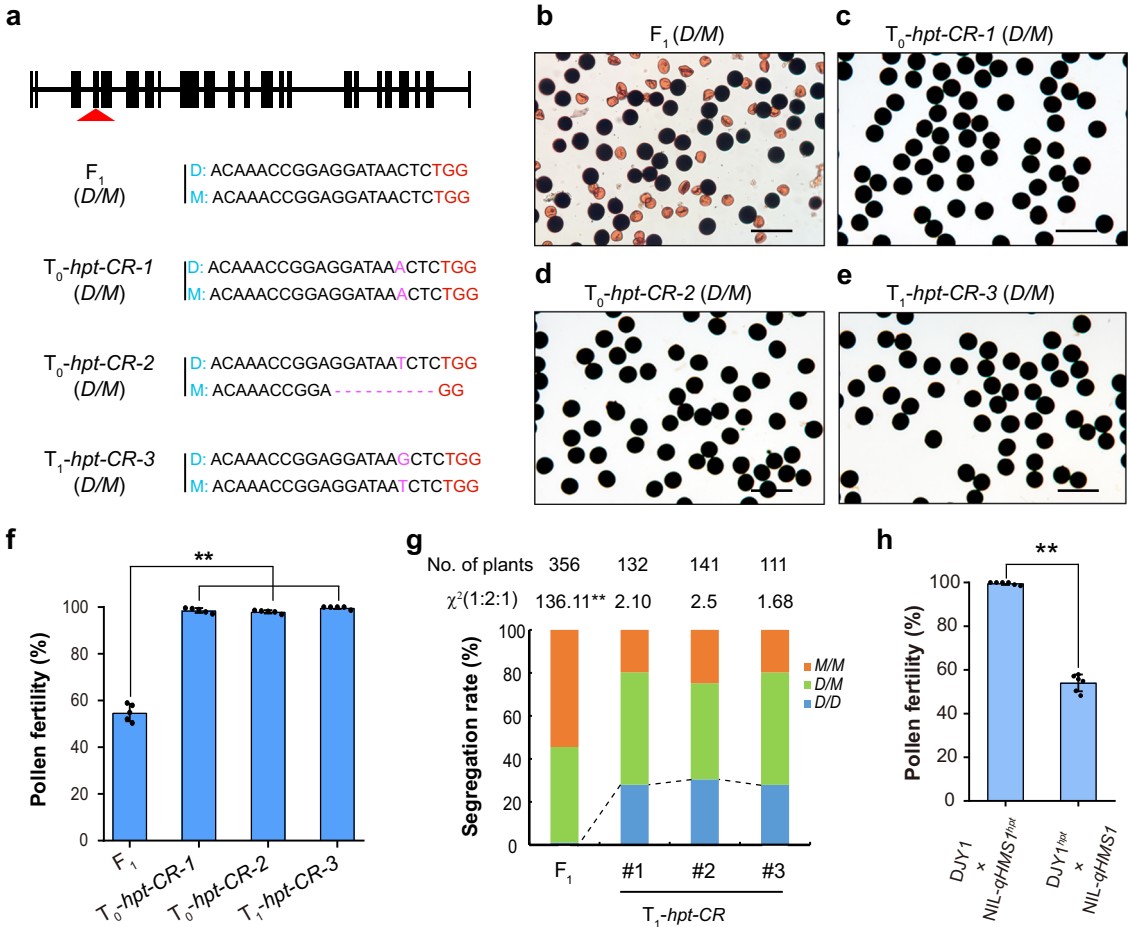

**Fig. 2 | HPT functions as a toxin. a** Creation of edited $F_1$ (DJY1/NIL-qHMS1) plants. The target site is indicated by red arrowhead. The PAM (TGG) is highlighted by red, and inserted bases by pink and deletion by pink dotted line. Two primary ($T_0$) and one progeny ($T_1$) plants are presented. **b–e** Pollen semi-sterility in transgene-negative $F_1$ (**b**) and restored pollen fertility in edited $T_0$ (**c, d**) and $T_1$ (**e**) plants. Scale bar, 100 μm. **f** Quantification of pollen fertility in plants shown in (**a**). Data are means ± SD ($n$ = 5 independent spikelets). **P = 1.60116E-5 < 0.01 by two-tailed

student's $t$ test. **g** Segregation of genotypes at qHMS1 fits into the 1:2:1 ratio upon knocking-out of HPT. **$P ≈ 0 < 0.01$ in $\chi^2$ test. **h** Verification of HPT from Mer as a functional toxin. HPT was knocked out in DJY1 ($DJY1^{hpt}$) or NIL-qHMS1 (NIL-$qHMS1^{hpt}$) background and $F_1$ was made by crossing for pollen fertility test. Data are means ± SD ($n$ = 6 plants). **$P = 9.26256E-10 < 0.01$ by two-tailed student's $t$ test. Source data are provided as a Source Data file.

progeny showed that all the offspring homozygous for the DJY1 allele of qHMS1 carried the HPA transgene, suggesting the cis-acting nature of HPA (Fig. 3i, Supplementary Tables 3, 4). Collectively, those results indicate that HPT and HPA from Mer together constitute a selfish genetic element.

## Truncation of HPT and a transposon insertion in HPA leads to the non-functional DJY1 allele

HPT encodes a member of the helicase subfamily, containing two RRM (RNA recognition motif) domains located in the N terminal, a DEAD-like helicases c domain (DEXDc), a Helicase superfamily c-terminal (HELICc) domain and a coiled coil (CC). The single base change of G to T in DJY1 leads to an early stop codon, truncating HPT from 902 amino acids (aa) to 670 aa, with both the HELICc and CC domains removed (Fig. 1h, Supplementary Fig. 12a). RNA-seq analysis of anthers at S9-S11 (early microspore stage (S9), vacuolated microspore stage (S10) and bicellular stage (S11)) revealed similar read depths between NIL-qHMS1, DJY1 and their $F_1$ plants across HPT (Supplementary Fig. 13a). HPT transcript was detectable in all the sampled tissues collected from plants at the flowering stage, with the preferred expression in the anther; and further analysis in developing anthers showed higher expression of HPT starting from early uninucleate stage (S9) through maturation of the pollen (S13) (Supplementary Fig. 13b, c). The HPT

promoter regions from Mer and DJY1 have similar activity as shown using the luciferase (LUC) reporter gene (Supplementary Fig. 13d). HPT was localized in both nucleus and cytoplasm in the transient protoplast assays and the truncated version from DJY1 retains its subcellular localization (Supplementary Fig. 14). Those results suggest that it is the truncation event that destroys toxicity of HPT in DJY1. The requirement of the C-terminal for functioning of HPT, where the HELICc and CC domains reside, was verified with a non-functional HPT created by a frameshift mutation near the G-to-T mutation in DJY1/NIL-qHMS1 $F_1$ (Supplementary Fig. 15).

HPA encodes an F-Box protein of 361 aa, with the F-Box motif located at the N-terminal, having two exons in its genomic structure (Fig. 1h, Supplementary Fig. 12b). Profiling of Mer HPA expression in the NIL-qHMS1 background indicates much preferred transcript accumulation in anthers, which reaches a peak at anther development stages S9-S10, roughly equivalent to early and late uninucleate pollens, respectively (Supplementary Fig. 13e, f). The anther-preferred expression of HPA was verified using the GUS reporter gene driven by a 1.4-kb promoter fragment of Mer-type HPA (Supplementary Fig. 13g). HPA was exclusively localized in nucleus (Supplementary Fig. 14). Next we investigated whether the 1391-bp transposon inserted 11-bp upstream of the ATG start codon or the single base change of T to C (Asn to Thr at aa level), or both, impact on the function of HPA at the

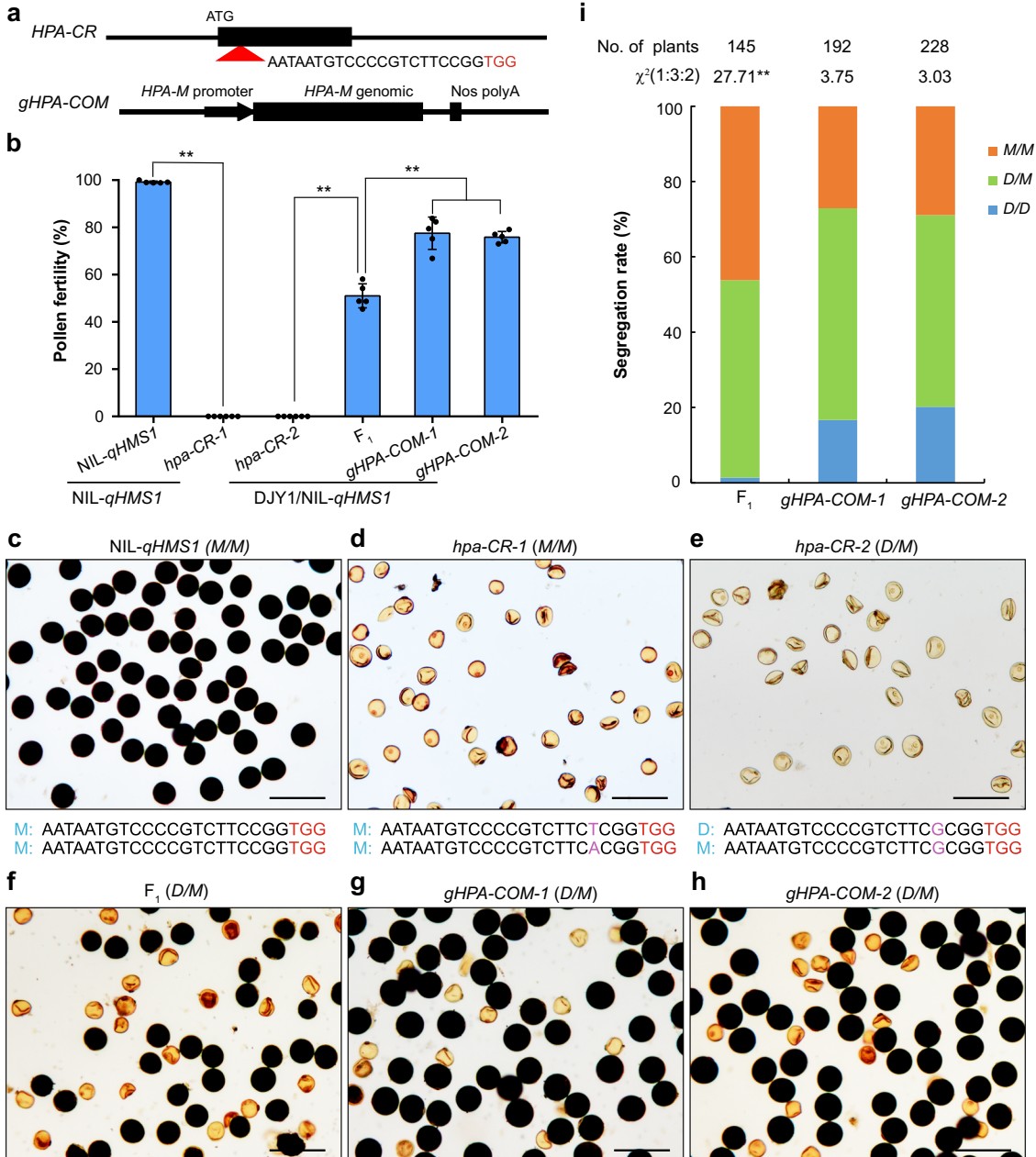

**Fig. 3 | HPA functions as an antidote. a** Target site in *HPA* (top) for CRISPR/Cas9-mediated mutagenesis and a *HPA* genomic fragment from *Mer* (bottom) for genetic complementation test. PAM is highlighted in red. **b** Pollen fertility in knock-out (*hpa-CR*) and complementation (*gHPA-COM*) lines. Data are means ± SD (*n* = 5 independent spikelets). **P* = 1.21003E-10 of NIL-*qHMS1* vs *hpa-CR-1*, 2.27263E-5 of F$_1$ vs *hpa-CR-2*, 0.00075 of F$_1$ vs *gHPA-COMs* < 0.01 by two-tailed student's *t* test. **c–h** Representative images of pollen phenotype in wild type (WT) (**c**) and knock-out (**d**) of NIL-*qHMS1*, and knock-out (**e**), WT (**f**) and complemented (**g**, **h**) lines of F$_1$ (DJY1/ NIL-*qHMS1*). Scale bar, 100 μm. **i** Observed segregation rate among progeny of the complemented lines, showing successful transmission of the DJY1 allele of *qHMS1*. **P* = 9.613E-7 < 0.01 in χ$^2$ test. Source data are provided as a Source Data file.

DJY1 allele (Fig. 1h). We performed RNA-seq analysis on anthers of NIL-*qHMS1*, DJY1 and their F$_1$, and found no transcript reads corresponding to the *HPA* coding region only in DJY1, implying that the transposon insertion disrupts transcription of *HPA* (Fig. 4a). The insertion-associated promoter inactivation was verified in the protoplast transient assay using LUC as a reporter (Fig. 4b). Further, RNA in situ hybridization on floral sections detected signal in NIL-*qHMS1* but not or faint signal in DJY1 (Supplementary Fig. 16). The promoter swapping experiment between DJY1 and *Mer* confirmed functionality of the DJY1-type HPA coding sequence, regardless of the Asn to Thr Amino acid substitution (Fig. 4c–e). Thus, it is the transposon insertion rather than

the single base substitution that leads to no accumulation of the antidote in pollens carrying the DJY1 allele of *qHMS1*.

To detect whether HPA directly interacts with HPT, we performed yeast two-hybrid and in vitro pull-down assays. The results showed no physical interaction between these two proteins (Supplementary Fig. 17). Helicase subfamily genes have previously been reported to perform diverse cellular functions in virtually all steps of RNA metabolism from bacteria to humans. With this knowledge, we performed RNA-seq experiments on anthers from the *gHPT-COM* transgenic plant carrying the *HPT-Mer* genetic fragment and displaying full pollen sterility in the DJY1 background, and DJY1 as a control (Supplementary

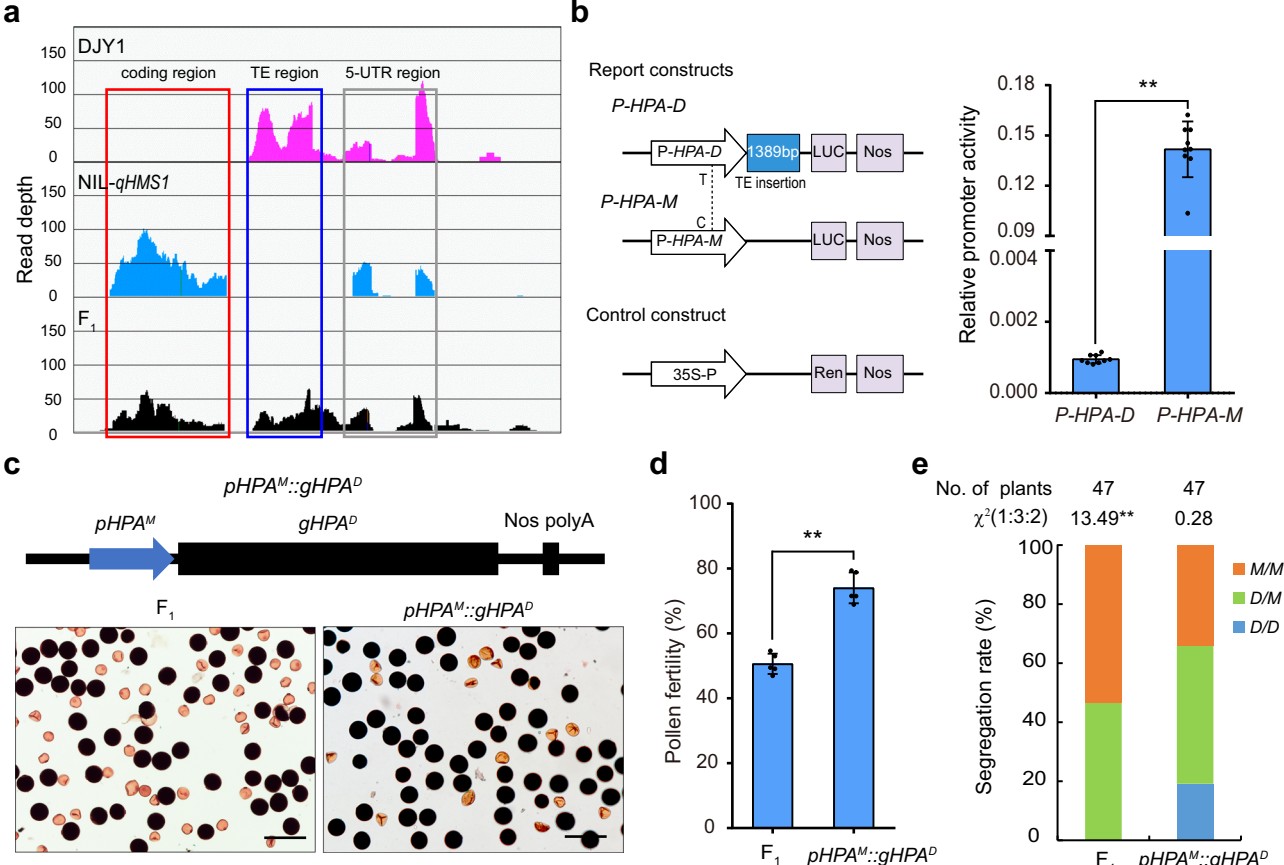

**Fig. 4 | A transposon insertion in *HPA* promoter region inactivates its transcriptional activity. a** RNA-seq analysis did not detect transcription from the coding region of *HPA* in DJY1 whereas *HPA* was transcribed in NIL-*qHMS1* and their F₁. TE region, transposon insertion region. **b** Promoter activity comparison of *HPA* between DJY1 and *Mer*. The promoter region from DJY1 (*P-HPA-D*, containing the transposon) or *Mer* (*P-HPA-M*) were used to drive the firefly luciferase (LUC) reporter gene in the protoplast transient assay. The Renilla luciferase gene driven by 35 S promoter was used as an internal control for data normalization. Data are means ± SD (*n* = 9 biologically independent samples). **P = 6.10922E-09 < 0.01 by

two-tailed student's *t* test. **c** The detoxification role of *HPA* from DJY1 is restored upon being transcribed. The DJY1 *HPA* promoter was replaced with the one from *Mer* and the resulting vector (*pHPAᴹ::gHPAᴰ*) was transformed into F₁ (DJY1/NIL-*qHMS1*). Representative images show partial restoration of pollen fertility. Scale bar, 100 μm. **d** Quantification of pollen fertility in *pHPAᴹ::gHPAᴰ* plants. Data are means ± SD (*n* = 4 independent spikelets). **P = 0.00053 < 0.01 by two-tailed student's *t* test. **e** Segregation of genotypes at *qHMS1* among progeny of a single-copy *pHPAᴹ::gHPAᴰ* plant. **P = 0.001177 < 0.01 in χ² test. Source data are provided as a Source Data file.

Fig. 18a, b). The anthers were sampled at the meiotic stage (S8) and the uninucleate stage (S9-S10) at which the first mitosis was arrested (Supplementary Fig. 6). The results showed that there were 3520 differentially expressed genes (DEGs) at the meiotic stage, with 2103 down-regulated and 1417 up-regulated (Supplementary Fig. 18c). Among the 2103 DEGs, 61 were down-regulated by 20 folds, of which 21 are anther-specific expressed (Supplementary Data 1; http://rapdb.dna.affrc.go.jp). At the uninucleate stage, there were 4905 DEGs, with 2717 down-regulated and 2188 up-regulated (Supplementary Fig. 18d). Among the 2717 DEGs, 20 were down-regulated by 20 folds, of which 7 are anther-specific expressed (Supplementary Data 1). Then we combined data from the two stages and identified 1953 co-DEGs, with 1243 down-regulated and 710 up-regulated (Supplementary Fig. 18e, f). GO analysis on those 1243 genes indicated that they were clustered in various pathways and cellular processes, particularly in the processing and metabolism of ncRNA and rRNA (Supplementary Fig. 18g). KEGG analysis on the same DEGs revealed various metabolic pathways, especially in the sugar and starch metabolisms (Supplementary Figs. 18h, 19). We performed qRT-PCR on some of those glycolysis or gluconeogenesis-related DEGs and confirmed their down-regulated expression in the *gHPT-COM* transgenic plant (Supplementary Fig. 20, Supplementary Table 5). Together, those results suggest a possible role of HPT in interfering expression of sugar/starch metabolic genes,

which in turn may affect subsequent starch granule formation during pollen development, leading to final pollen abortion.

## *qHMS1* has evolved non-functional alleles in cultivated rice and its closest wild relatives

Using 100 wild rice and 244 landrace varieties (Supplementary Data 2), we traced the evolutionary trajectory of the *HPT* and *HPA* from *Mer* to cultivated rice. For *HPT*, the genomic sequence is identical in earlier wild species of the AA genomes group, but changes were found in *O. barthii*, the wild progenitor of cultivated rice in Africa, and in *O. rufipogon*, the wild progenitor of cultivated rice in Asia (Fig. 5a, Supplementary Fig. 21a). *HPT* is lost in all the *O. barthii* accessions and cultivated African landraces (*O. glaberrima*) analyzed. *HPT* is diversified in *O. rufipogon*, with majority of accessions (53 out of 76) retaining the *Mer* version of *HPT* and minority (23 out of 76) having evolved the DJY1 version (with early stop due to the G to T mutation). In 233 descendants (*O. sativa*) of *O. rufipogon*, 67.8% landraces carry *Mer HPT* and 32.2% carry DJY1 *HPT* (Fig. 5a). Thus, loss-of-function of *HPT* is a recent evolutionary event. The similar frequencies of *Mer HPT* and DJY1 *HPT* between *O. rufipogon* and *O. sativa* suggest that *HPT* was not selected during domestication of cultivated rice.

For *HPA*, its protein coding region remains unchanged in all the analyzed accessions, including wild and cultivated rice, except the

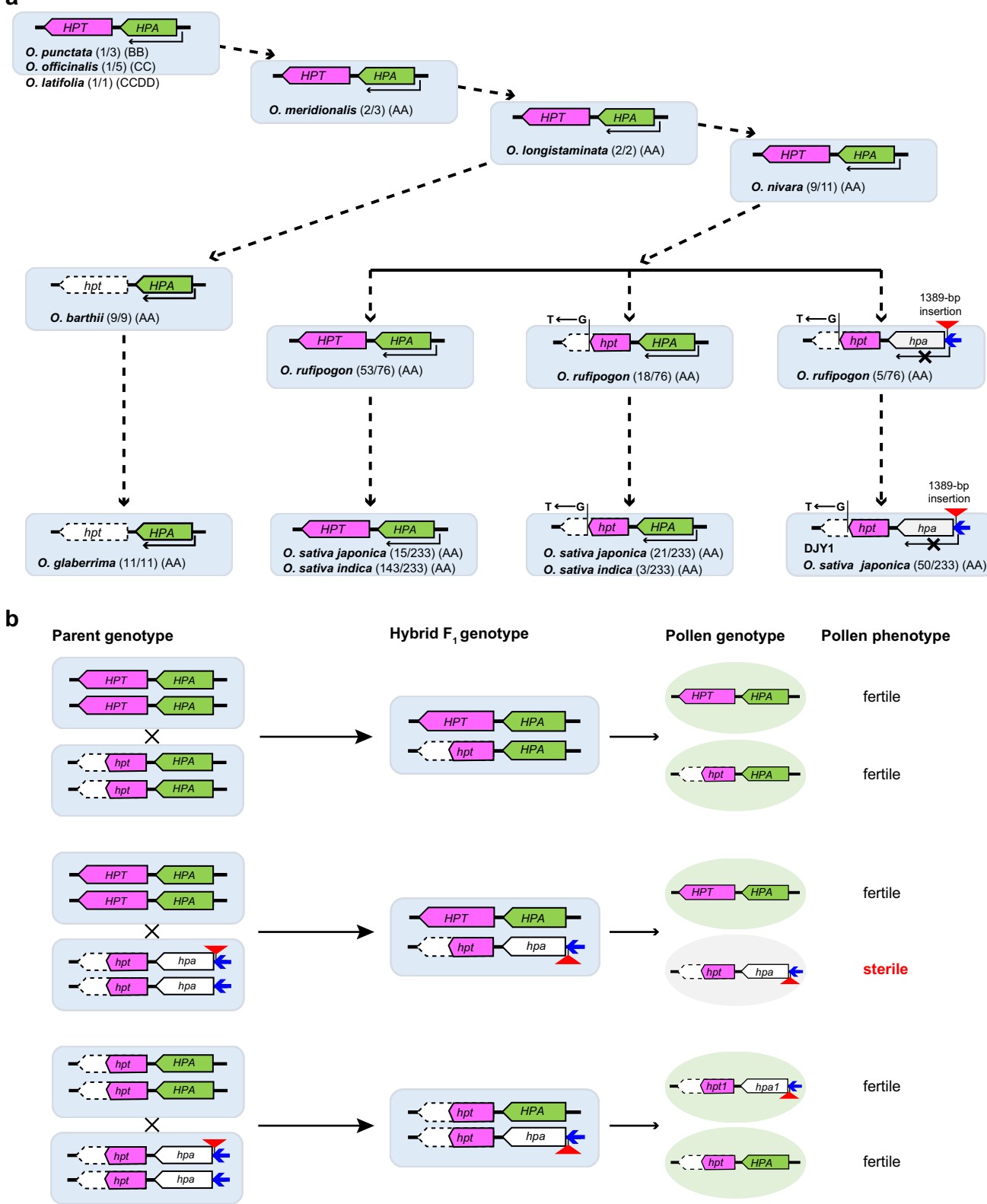

**Fig. 5 | Loss-of-function of *qHMS1* is a recent evolutionary event. a** Coupling of the toxin HPT with the antidote HPA. Mutation of *HPT* and *HPA* was found in cultivated rice and its closest progenitor in Asia and Africa. Mutation of *HPA* is conditioned by mutated *HPT*. Loss or truncation of HPT is indicated by white bars with dotted line. Inactivation of *HPA* transcription due to the transposon insertion is indicated by white bars with black line. Shown in parenthesis is number of a given genotype over total accessions analyzed. **b** Predicted hybrid pollen sterility in combinations between the two *qHMS1* alleles as a guide in breeding programs. The varieties with mutated *HPT* but functional *HPA* are considered having wide compatibility. The combination that produces pollen sterility is highlighted in red rectangle.

single T to C mutation that has no impact on function of *HPA* as described above (Fig. 4c–e). The transposon insertion in the *HPA* promoter region exists in 6.6% (5/76) *O. rufipogon* (Supplementary Fig. 21b) and 21.9% (51/233) *O. sativa* accessions, particularly only in some (51/87) of *O. sativa ssp. japonica* rice (Fig. 5a). Interestingly, *HPA* remains intact in African rice where *HPT* no longer exists. It is likely that the insertion event occurred after the loss-of-function mutation of *HPT* as the site-specific insertion in *HPA* does not exist in the context of a functional *HPT*, or any loss-of-function mutation of *HPA* would not survive at the presence of *Mer HPT* (Fig. 5a). The fact that the functional *HPT* must be coupled with the expressed *HPA* is also true in recently sequenced 251 and 33 varieties representing diverse genetic backgrounds[39,40] (Supplementary Data 2), which at least partially explains why the silenced *HPA* exists at a lower frequency. We constructed a phylogenetic tree based on *HPA* sequences in the eight rice species of the AA genome group. This tree shows again the co-evolution of silenced *HPA* with the non-functional allele of *HPT* (Supplementary Fig. 22). Accordingly, we propose a model that would provide a guide in choosing parents for hybrid breeding in consideration of pollen fertility (Fig. 5b). An extended protein sequence analysis to more distant species, including wheat and sorghum, suggests possible conservation of these two proteins, particularly the

F-Box in HPA and DEXD and HELICc domains in HPT (Supplementary Figs. 23, 24, Supplementary Tables 6, 7). Whether those distant versions have function in maintaining species identity remains to be elucidated.

## qHMS1 provides the first tier of hybrid sterility in a genetic stack with qHMS7

The newly evolved functional DJY1 allele at the *qHMS7* locus kills the pollen carrying the non-functional *Mer* allele[31], opposite to the *qHMS1* locus where the functional *Mer* allele kills the pollen carrying the newly evolved non-functional DJY1 allele as shown in this study. We therefore brought those two loci together by crossing NIL-*qHMS1* with NIL-*qHMS7*, both in the DJY1 background and investigated pollen fertility in the F$_1$ plant. We observed ~75% pollen sterility, being 50% earlier abortion (at the uninucleate stage, associated with the DJY1 allele of *qHMS1*) and 25% later abortion (at the binucleate stage, associated with the *Mer* allele of *qHMS7*), indicating that half of the 50% alive pollens (carrying the *Mer* allele of *qHMS1*) were later killed by the DJY1 toxin-antidote system of *qHMS7* (Fig. 6a–e). Consequently, nearly all the F$_2$ progeny contain at least one functional copy of the two toxin-antidote systems, thus verifying selective transmission of the pollen carrying both the *Mer* allele of *qHMS1* and the DJY1 allele of *qHMS7* (Fig. 6f–g).

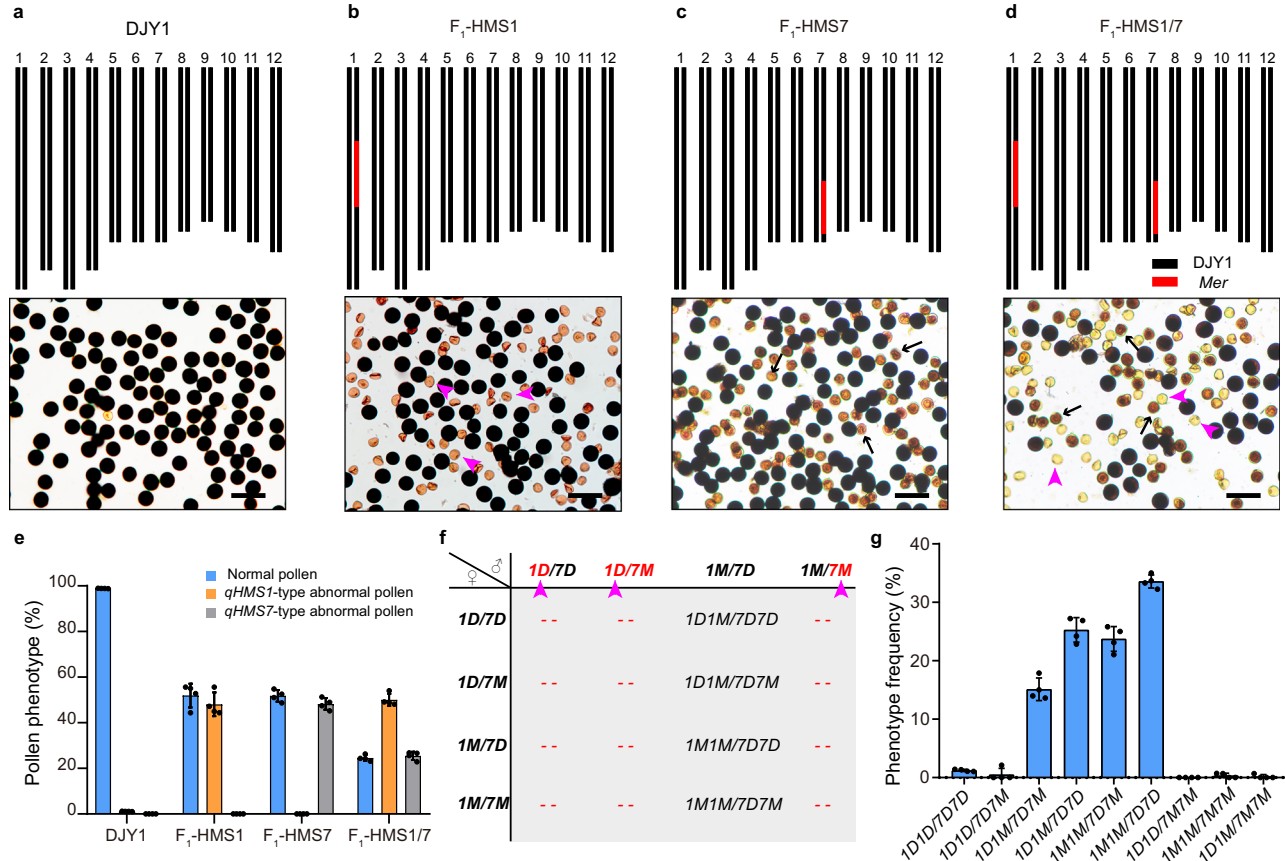

**Fig. 6 | *qHMS1* and *qHMS7* are independent in action and together provide tiered hybrid male sterility. a–d** Chromosome location of *qHMS1* and *qHMS7* and representative images of pollen phenotype in DJY1 (**a**), DJY1/*qHMS1* F$_1$ (F$_1$-HMS1) (**b**), DJY1/*qHMS7* F$_1$ (F$_1$-HMS7) (**c**) and *qHMS1*/*qHMS7* F$_1$ (F$_1$-HMS1/7) (**d**). The pollen abortion caused by *qHMS1* at uninucleate stage and that caused by *qHMS7* at binucleate stage are indicated by pink arrowheads and black arrows, respectively. Scale bar, 100 μm. **e** Quantification of pollen phenotypes in hybrid plants. Data are means ± SD (*n* = 5 independent spikelets). **f** Expected genotypes among progeny of F$_1$-HMS1/7. Pink arrowheads indicate the aborted pollen. *1D*, DJY1 allele of *qHMS1*; *1 M*, *Mer* allele of *qHMS1*; *7D*, DJY1 allele of *qHMS7*; *7 M*, *Mer* allele of *qHMS7*.

**g** Genotypic distribution among progeny of F$_1$-HMS1/7 plants (*n* = 4 F$_2$ populations from four randomly chosen F$_1$ plants). The prevalence of the four genotypes verifies the expectation in **f**. Note the uneven distribution frequency of the four genotypes. The female gametes carrying both the functional TAs (1 M/7D) have the highest transmission whereas those carrying both the none-functional TAs (1D/7 M) have the lowest transmission, suggesting that two TAs together can build up partial hybrid female sterility. Four F$_2$ populations (100–200 plants each) from 4 randomly chosen F$_1$ plants were genotyped. Data are means ± SD. Source data are provided as a Source Data file.

Since only one genotype pollen (*1M/7D*, denoting *qHMS1-Mer/qHMS7-DJY1*) in the F$_1$ plants heterozygous at both *qHMS1* and *qHMS7* is viable, four genotypes (*1D1M/7D7M*, *1M1M/7D7M*, *1D1M/7D7D* and *1M1M/7D7D*) are expected at a 1:1:1:1 ratio in the F$_2$ population if the four female gametes (*1D/7M*, *1M/7M*, *1D/7D* and *1M/7D*) are transmitted equally (Fig. 6f). However, we observed the highest frequency of the *1M1M/7D7D* genotype and the lowest frequency of the *1D1M/7D7M* genotype, biased from the expected equal segregation (Fig. 6g). This result implies higher transmission rate of the female *1M/7D* gamete, suggesting that *qHMS1* and *qHMS7* together can lead to partial female gamete sterility although they each alone does not do so. Those results demonstrate that the two toxin-antidote systems are independent of each other in action and together can confer tiered reproductive isolation.

## Discussion

Understanding the genetic mechanism of hybrid male sterility in rice is not only a central issue in evolution biology, but also has directly important applications in breeding. Most of the causal genes responsible for hybrid male or female sterility were identified in the studies aimed to utilization of robust heterosis between subspecies of cultivated rice. Knocking out of some of those genes in either male or female parent has been recently shown to restore hybrid fertility to almost a practical level in seed production[41]. Wild rice species constitute important gene pools that can be exploited to develop new rice varieties with resistance to biotic and abiotic stresses and to challenge global climate changes. Cloning of genes governing hybrid sterility between wild and cultivated rice, followed by targeted mutagenesis to those genes may allow efficient gene flow from wild gene pools to modern cultivars. However, few studies have been focused on interspecific reproductive isolation. We started to work on hybrid male sterility between *Mer* (the most distant wild species in the AA genome group) and cultivated rice over a decade ago. In this study, we have identified causal genes at *qHMS1*, as an addition to *qHMS7* cloned previously[31], thus moving one more step towards cloning of all the four mapped loci.

Toxin-antidote systems are common in nature although the involved genes and encoded proteins are diverse among them. For example, the toxin-antidote element was first described as a single locus in the beetle Tribolium[42]. Cloning of the genes *peel-1* and *zeel-1*, encoding a naturally occurring sperm-derived toxin (a four-pass transmembrane protein) and antidote (a six-pass transmembrane protein), respectively, was reported in nematodes more than a decade ago[16,17]. Along with the discovery of more toxin-antidote systems in *C. elegans*[18] and *C. tropicalis*[43], analogous effects between them were found, in which those elements act independently and when combined together, confer extensive genetic incompatibility. Although only a few have been reported in plants, including the one recently described in Arabidopsis[25], the selfish genetic elements seem common in nature and likely play key role in speciation and maintenance of species identity. Different from those found in some other species, where a single gene can encode both toxin and antidote proteins, such as the *wtf* genes in *saccharomyces pombe*[14,15], the *Sk-1* genes in *neurospora*[24] and the *Spok* and *Het-s* genes in *Podospora anserina*[22,23], plant selfish genetic elements are usually encoded by at least two genes. Interestingly, both the toxin-antidote systems at *qHMS1* and *qHMS7* are encoded by two linked genes. Whether this is specific to hybrid between *Mer* and *O. sativa* remains to be investigated in future studies.

The proteins encoded by *qHMS1* are completely different from those by *qHMS7*. HPT and HPA belong to DEAD-like RNA helicases and F-box protein, respectively, whereas the toxin and antidote encoded by *qHMS7* are RIP domain-containing protein and grass family-specific protein[31], respectively. In addition, they also have different subcellular localizations (HPT in cytoplasm and nucleus and HPA in nucleus vs ORF2 in cytoplasm and nucleus and ORF3 in mitochondria). Regards of the timeline for pollen abortion, it is likely that *qHMS1* exerts its effects prior to the action of *qHMS7*. It is unknown if the protein features and localizations determine their action order. Moreover, the malfunctionization of *qHMS1* in the latest species has clear simple path and more recent. In contrast, the evolution trajectory of functionization of *qHMS7* is a long path and much more complicated[31]. However, we cannot exclude the possibility that the formation of functional *qHMS1* had a long evolution trajectory before *Mer*.

The DEAD-box helicases are a large family of conserved RNA-binding proteins that belong to the broader group of cellular DExD/H helicases and perform diverse cellular functions in virtually all steps of RNA metabolism from bacteria to humans[44–46]. F-box proteins function as substrate adaptors for the SCF ubiquitin ligase complexes, which mediate the proteasomal degradation of a diverse range of regulatory proteins[47]. The F-box protein-involved self-incompatibility/compatibility has been reported[48–51]. We hypothesize that HPT interferes RNA metabolisms related to early pollen development and HPA detoxifies HPT possibly through SKP1-CULLIN1-F-box (SCF)-mediated protein degradation. However, our in vitro assays did not detect their physical interaction, implying that the HPA-associated SCF complex may work on a substrate that in turn regulates HPT. On the other hand, HPT is synthesized in the cytoplasm and may be transported to the nucleus (in consistence with its subcellular localization), where it interferes with RNA metabolism. When nucleus-localized HPA is available, the HPT degradation process is triggered on, thus protecting pollens from abortion.

HPT acts in a sporophytic manner whereas HPA in a gametophytic manner. It is possible that the cytoplasm-localized HPT (possibly shuttling in and out of nucleus) but not the nucleus-localized HPA in the pollen mother cell is transmitted through meiosis to the microspore, such that pollens lacking HPA fail to develop due to the presence of the toxin HPT. A recent study in maize establishes that gametophyte genome activation occurs at pollen mitosis I[52]. We therefore also postulate that the *HPT* transcript transcribed in the pollen mother cell may be more stable relative to that of *HPA*, survive meiosis until the microspore uninucleate stage and then be translated, killing pollens without HPA, as pollen abortion starts to be seen at the uninucleate stage (Supplementary Figs. 4, 5). Why HPT is selectively toxic to male gametophyte, as also seen in other toxin-antidote systems[31,32], remains mysterious. Nevertheless, we have decoded another quantitative locus, in addition to *qHMS7*, in our endeavor to understand hybrid sterility control between the earliest wild rice and cultivated rice in the AA genome group. As approaching to clone other the two mapped loci[31], we expect that knocking out multiple hybrid sterility genes would provide interesting insight into breaking reproductive isolation formed during million years evolution, to finally realize more efficient gene flow from wild to cultivated rice.

## Methods

### Plant material and growth condition

The wild rice *Oryza meridionalis* (accession 104498, *Mer*) and the *O. sativa ssp. japonica* cultivar Dianjingyou1 (DJY1) were used to produce F$_1$. The F$_1$ plant was then backcrossed for 5 times using DJY1 as the recurrent parent and the resulting BC$_5$F$_1$ was self-pollinated to finally generate the near-isogenic line NIL-*qHMS1*. Genetic background of NIL-*qHMS1* was scanned using 453 simple sequence repeat (SSR) markers evenly distributed on the 12 chromosomes (Supplementary Fig. 1 and Supplementary Data 3). All the wild rice and cultivated rice species used for evolutionary analysis were either maintained in our lab or provided by the Chinese Crop Germplasm Resources Center (Beijing, China). All plants were grown in field at our experimental stations in Nanjing or Beijing during the summer season, or in Sanya, South China during winter season.

## Evaluation of pollen fertility

Spikelets collected from plants at the flowering stage were fixed in Carnoy's fixative (100% ethanol: 100% acetic acid = 3:1). Pollen grains from the fixed spikelets were stained by 1% (w/v) iodine-potassium iodide ($I_2$-KI) solution. At least three spikelets (~ 200 pollens counted each) were examined for pollen fertility under a light microscope (Leica DM5000 B). Dark-stained pollen grains were scored as fertile, whereas irregular-shaped, unstained grains were scored as sterile. Pollen fertility is measured by Image J software.

## Cytological analyses

The developmental course of anthers was classified into 14 stages[53,54]. For semi-thin sections, anthers of different developmental stages were first fixed in Carnoy's fixative, underwent a series of ethanol dehydration and resin penetration, and then were embedded in resin (Technovit 7100, Hereaus Kulzer). Sections were prepared using an EM UC7 Ultramicrotome (Leica), adhered to a glass slide, and then stained with 0.25% toluidine blue O (Chroma Gesellschaft Shaud). Images were captured using a light microscope (Leica DM5000 B). For aceto-carmine staining, the microspores were released onto the slide by breaking anthers fixed in Carnoy's fixative and stained with 1% (w/v) aceto-carmine solution. two minutes later, the samples were examined under a light microscope. The observation was repeated 3 times (one spikelet per replicate) for each developmental stage to calculate the frequency of microspores (or pollen grains) of different types. The procedure of DAPI (4′, 6-diamidino-2-phenylindole) staining was basically the same as for the aceto-carmine staining, except that the dye was replaced with DAPI and glycerin. The samples were examined under a fluorescence microscope (Leica DM5000 B). For SEM and TEM analysis, spikelets at the mature stage were fixed in 2.5% (v/v) glutaraldehyde for 24 h to 72 h, post-fixed by 1% $OsO_4$ in PBS, pH 7.2. After dehydrating through an ethanol series, samples were placed in Spurr's resin and then ultrathin sectioned. Furthermore, sections were double-stained with 2% (w/v) uranyl acetate and 2.6% (w/v) lead citrate aqueous solution, and examined with a Jeol 100 CX electron microscope (Jeol Ltd, Tokyo, Japan).

## Fine mapping of the *qHMS1* locus

The *qHMS1* locus was first delimited to a genomic interval between the markers CH1-11 and RM3475 using 356 $BC_5F_2$ and 8 molecular markers. A series of Indel markers were designed based on genomic sequence difference nearby the mapped region between *Oryza meridionalis* and Nipponbare (*O. sativa, japonica*) at the Gramene website (http://www.gramene.org/). The SNP markers were based on sequences of site-specific PCR products amplified from DJY1 and NIL-*qHMS1*, respectively. For fine mapping, 11,800 plants from the progeny of the $BC_5F_2$ population were screened for recombinants using the two Indel makers (E1-3 and E1-6). Finally, the *qHMS1* locus was delimited to a 43-kb genomic region between the SNP markers C1-20 and C1-39. All the primers for mapping are listed in Supplementary Data 4.

## RNA isolation and quantitative reverse transcription PCR analysis

Various tissues at the flowering stage and anthers at defined development stages were collected from DJY1, NIL-*qHMS1* and $F_1$ plants and stored at −80 °C. Total RNA was extracted using an RNA Prep Pure Plant Kit (TIANGEN). The first strand cDNA was synthesized using Prime Scriptase (TaKaRa). Quantitative real-time RT-PCR, with 3 biological replicates, was performed using SYBR Premix Ex Taq II (TaKaRa) on an Applied Biosystems 7500 Real-Time PCR System. Amplifications were performed at 95 °C for 5 s, 60 °C for 34 s, 40 cycles. The *UBQUITIN1* (*LOC_OsO3g13170*) gene was used as an internal control. All the primers used in this analysis are listed in Supplementary Data 4.

## RNA-Sequence analysis

The spikelets of DJY1, NIL-*qHMS1,* and $F_1$ were collected at stages 9 to 11 and the anthers of *gHPT-COM* and DJY1 were collected at the meiotic stage and the uninucleate stage. Total RNA of different tissues was extracted according to the above method and RNA-Sequencing libraries were prepared with three biological replicates. Then, the libraries were sequenced separately using Illunima HiSeq X-10 platform (Biomarker Biotechnology Co) and generated reads were cleaned and then mapped to the Nipponbare reference genome using the HISAT[55] and Bowtie2[56] tools. RNA-Sequence data was browsed using IGV software.

## CRISPR/Cas9 vector construction

The 20-bp target-specific spacer sequences were synthesized, fused with the AarI linearized intermediate vector SK-Grna and introduced into CRISPR-Cas9 binary vector pCAMBIA1305 to generate knock-out constructs[57]. Then, these vectors were transformed into the callus of DJY1, NIL-*qHMS1* or $F_1$ via an Agrobacterium tumefaciens–mediated transformation system[58]. Transgenic plants were regenerated from transformed callus by selection on hygromycin-containing medium. The edited types of transformants were identified by sequencing. Genes sequences were analyzed using Geneious software. Primers used in the plasmid construction are listed in Supplementary Data 4.

## Genetic complementation

The full-length genomic fragments of corresponding genes were obtained from DJY1 and NIL-*qHMS1* by PCR amplification. Then, these fragments were fused into the vector pCUbi1390 using the In-Fusion Advantage PCR Cloning Kit (Clontech) to generate 1390-*HPT*-D, 1390-*HPT*-M, and 1390-*HPA-M* genetic complementation vectors. These vectors were transformed into the callus of DJY1, NIL-*qHMS1* or $F_1$ via an Agrobacterium mediated transformation system. Positive transgenic individuals were identified through PCR using transgene-specific primers. All primers used for plasmid construction are listed in Supplementary Data 4.

## Subcellular localization

The full-length CDSs of HPT and HPA without stop codons were fused with a green fluorescent protein (GFP) and inserted into the pAN580-GFP vector to produce pAN590-3-D-GFP, pAN590-3-M-GFP and pAN590-5-M-GFP constructs. Rice protoplasts was prepared as the following steps: the stems of 9311 seedlings, which had been growing for 7–10 days, were cut into small segments of about 1–2 mm and placed in 10 ml of 0.6 M mannitol for 10 min. Remove mannitol, add 10 ml enzyme solution (activated at 42 °C), shake at 25 °C at 60 rpm, enzymatic hydrolysis for 5 h. Remove the enzyme solution, add 30 ml W5 solution in two batches, shake gently, pass through a 200-mesh sieve, and collect the filtrate. Centrifuge at 1000 rpm for 3 min and remove the supernatant. Rinse with 4 ml W5 solution once, centrifuge and remove supernatant. The protoplasts were obtained by adding 2 ml MMG solution for re-suspension[59]. Plasmid DNA of sequence-confirmed pAN580-GFP and the nuclear marker D53-mCherry (as control) were co-transformed into rice protoplasts and incubated in darkness for 16 h at room temperature. Subcellular distribution of GFP fluorescence was visualized using a confocal laser scanning microscope (Zeiss LSM 780). All the primers used here are listed in Supplementary Data 4.

## Transactivation assays

An approximately 2.5-kb promoter region of *HPT-D* and *HPT-M*, 2.7-kb promoter region of *HPA-D* and 1.6-kb promoter region of *HPA-M* were amplified from DJY1 or NIL-*qHMS1* genomic DNA by PCR and cloned into the pGreenII0800-LUC vector to generate P-*HPT*-D, P-*HPT*-M, P-*HPA*-D and P-*HPA*-M LUC reporter constructs. Sequencing confirmed

plasmids DNA were transformed into rice protoplasts[59]. The *luciferase* gene from *Renilla reniformis* (Ren) under control of the CaMV35S promoter was used as the internal control. The LUC activity was calculated with a Dual-Luciferase Assay Kit (Promega E2920) following the manufacturer's recommendations using the TriStar2 Multimode Reader LB942 (Berthold Technologies) and the relative LUC activity was represented by the ratio of LUC/Ren. All the primers used here are listed in Supplementary Data 4.

### β-Glucuronidase (GUS) histochemical staining

An approximately 1.6-kb promoter fragment upstream of the start codon of *HPA-M* were amplified from NIL-*qHMS1* genomic DNA by PCR and cloned into the binary vector pCAMBIA1381 to drive GUS reporter gene expression. Then, this vector was transformed into the callus of NIL-*qHMS1* via an Agrobacterium tumefaciens–mediated transformation system. Young spikelets from positive T$_0$ plants were stained with GUS dye and decolorized with 75% alcohol[60]. Images were captured using a light microscope (Leica DM5000 B). The primers for PCR amplification are listed in Supplementary Data 4.

### Yeast two-hybrid (Y2H) assays

Y2H assays were performed using the Matchmaker Two-Hybrid System (Clontech, CA, USA) following the manufacturer's protocol. The coding sequence of *HPA* was amplified and cloned into the pGBKT7 vector (bait), while the *HPT* was cloned into the pGADT7 vector (prey). These two constructs were co-transformed into yeast strain AH109 and selected on medium plates without Leu and Trp (SD/-Leu-Trp) at 30 °C for 3–5 days. The interactions were tested on selective medium (SD/-Leu-Trp-His) and (SD/-Leu-Trp-His-Ade) plates. Primers used in the assays are listed in Supplementary Data 4.

### Protein pull-down assay

The coding sequence of *HPA* was introduced into the pMAL-C2x vector to construct the fusion protein MBP-HPA, and *HPT* was cloned into the pGEX-4T-2 vector to make the GST-HPT fusion protein (primers used in the assays are listed in Supplementary Data 4). The proteins including fusions and empty tags were expressed in Escherichia coli DE3 cells (TransGen). Different combinations of proteins expressed in vitro were mixed together and incubated with GST-beads overnight[61]. The pull-down assays were performed and samples were detected with anti-GST direct antibody (WBL) or anti-maltose-binding protein (anti-MBP) (NEB) monoclonal antibody at 1: 5000 dilution, and the secondary antibody was anti-IgG (WBL) mouse (1: 5000 dilution; Abmart). The ECL reagent (Bio-Rad) was used to test the immunoblot.

### RNA in situ hybridization

Young spikelets from DJY1 and NIL-*qHMS1* were fixed in a FAA (RNase free) fixative solution at 4°C. Through dehydration in a series of ethanol and xylene, simples were embedded in paraffin (Paraplast Plus, Sigma). A probe was prepared by amplifying a fragment from HPA-M cDNA and cloned into the pGEMT Easy vector (Promega). The probe was then transcribed in vitro using a DIG Northern Starker Kit (Cat. no. 2039672, Roche) following the manufacturer's instructions. RNA in situ hybridization with the probes were performed on transverse sections of anther. After blotting with Antidigoxigenin AP-conjugated (Roche, 11093274910) and incubation with the NBT solution (Roche, 11383213001), the slides were observed and photographed using a light microscope (Leica DM5000 B)[60]. Primers used are listed in Supplementary Data 4.

### Amino acid sequence alignment

The full-length amino acid sequence of HPT and HPA were used to search the homologous sequences in the National Center for Biotechnology Information (NCBI, http://www.ncbi.nlm.nih.gov/). Amino acid sequence alignment was constructed using Bioedit software.

Phylogenetic tree was constructed using MEGA-X software (neighbor-joining method with 1000 bootstrap replicates).

### Reporting summary

Further information on research design is available in the Nature Portfolio Reporting Summary linked to this article.

## Data availability

All data supporting the finding of this work are available within the paper and its Supplementary Information files. Plant materials generated in this study are available from the corresponding author upon request. Nipponbare reference genome is available at Gramene website (http://www.gramene.org/). Gene expressed profile data were collected from The Rice Annotation Project Database (http://rapdb.dna.affrc.go.jp). Genotypes of different varieties are collected from Rice Resource Center (http://ricerc.sicau.edu.cn/) and RiceSuperPIRdb (http://www.ricesuperpir.com/). The RNA-Seq data used in this study have been uploaded to NCBI under BioProject PRJNA958444, PRJNA959200 and PRJNA1009707. Source data are provided with this paper.

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

## Acknowledgements

This research was supported by National Natural Science Foundation of China (Grant No. 31991224 to X.Y., U2002202 to Z.Z., 31971909 to Z.Z., 31901480 to X.Y.), Jiangsu Research and Development Program (Grant No. BE2021360 to Z.Z.), the Natural Science Foundation of Jiangsu Province, China (Grant No. BK20200023 to X.Y.), the Fundamental Research Funds for the Central Universities (Grant No. ZJ22195020 to X.Y.) and the Jiangsu Collaborative Innovation Center for Modern Crop Production.

## Author contributions

S.Y., Z.Z. and X.Y. contributed equally. S.Y. and Z.Z. conceived the study and designed the experiments. X.Y. performed the genetic stack. J.Z.,

Jing Li. and D.T. constructed the near-isogenic line. Xiaoming Zheng provided wild and cultivated rice accessions. S.Z., Jian Wang, D.L., H.C., Y.X., W.C., Q.W., Jiayu Lu, K.C. and C.Z. provided technical support during the study. Xin Zhang, Z.C., X.G., Y.R., Shijia Liu., X.L., Y.T., and Ling Jiang participated in material development. Jianmin Wan supervised the study. S.Y. and Z.Z. drafted the manuscript with input from C.W. All authors contributed to subsequent versions and have read and approved the manuscript.

## Competing interests

The authors declare no competing interests.
