## [Peer Review File · Nature Communications]

A Toxin-Antidote System Contributes to Interspecific Reproductive Isolation in RiceReviewers' Comments:

Reviewer #1:

Remarks to the Author:

This work cloned a new hybrid male sterility locus, qHMS1, between *Oryza sativa* and *O. meridionalis* using map-based cloning. This locus contained two components, a toxin ORF3 and an antidote ORF5. Pollen grains without the antidote will be selectively killed in the hybrid. Such a working model is exactly similar to that of qHMS7, a previously identified hybrid male sterility locus. This work further investigated the origin of qHMS1 using 100 wild rice and 244 landrace varieties. Finally, they constructed stacked lines with both qHMS1 and qHMS7 and evaluated the effect on hybrid fertility. Generally, this work extends the understanding of hybrid sterility and has certain application significance in agriculture. Suggestions are listed below for consideration.

1. The novelty of the work is not well presented. For example, the author described "a mode of action like qHMS7". However, we are more interested in findings that are different from those of previous studies. I suggested the authors clearly indicate what's new in the study in mechanism or anything else.
2. The introduction might expand slightly to cover the progress in hybrid incompatibility in other plant species since there are not too many interspecific cases in rice.
3. Any information about the molecular markers mentioned in the result?
4. Extended Data Fig. 9. It seems that the Mer allele of ORF4 was not edited, as indicated in the figure. Is it possible that if the Mer allele of ORF4 was involved in qHMS1?
5. Any physical interaction between ORF3 and ORF5? How does ORF3 trigger abortion of male gametes? What is the possible role of HELICc and CC domains? Why does ORF5 restore fertility, and what is the pathway leading to protection of the gamete? RNA-seq might provide a possible explanation for the answers.
6. Extended Data Fig. 15. I am confused with the signal because there seems no difference between the antisense and sense probe for both DJY1 and NIL-qHMS1.
7. Figure 5. The authors might add a phylogenetic tree of the rice species to better present the evolutionary event of qHMS1. Is that mean that the loss-function of ORF3 occurred independently in the African rice and Asian rice? In addition, the arrows in Figure 5b are confusing, and this figure might be reorganized.
8. Line 199-202. It is not easy to understand the description.

Minor

1. Some descriptions in the introduction are not accurate. For example: The male HS loci qHMS7, Sa and Sc, female HS locus S5 and female-male HS locus S1 fit into this model. However, those genes explain HS between subspecies of *Oryza sativa* (with qHMS7 as an exception). However, S1 is responsible for hybrid sterility between Asian and African species.
2. Extended Data Fig. 1b, Genomic fragments from Mer identified by whole-genome scanning with molecular markers, any detail?
3. Extended Data Fig. 2. No description for (a).
4. Extended Data Fig. 3. In the figure legend, I guess "D, DJY1 allele; M, Mer allele" was associate with Figure 1.
5. Figure 1g. Please show the data of fertility of the recombinant.
6. Extended Data Fig. 9. Please provide the statistical analysis of the data.
7. Line 87-90. I suggested to describe the sequence variance of the five ORFs especially ORF3 (Line 116-120) and ORF5 (Line 142-144) here.

Reviewer #2:

Remarks to the Author:

In this manuscript, You and colleagues dissect the genes underlying a novel toxin-antidote element

responsible for hybrid male sterility in a cross between a wild and domesticated rice. Previously, the authors identified four major quantitative hybrid male sterility loci (qHMS1, qHMS2, qHMS7, and qHMS9) contributing to sterility of the F1 hybrid between *O. meridionalis* and *O. sativa* DJY1. In a recent study the authors found that qHMS7 contains a toxin-antidote element. In the present study, the authors extend this work and show that a second sterility locus, qHMS7 also contains a toxin-antidote element and that these two selfish elements act independently, as expected from the original cross. Understanding the genetic basis of male sterility in rice is not only interesting from an evolutionary point of view (speciation) but it has direct applications in industry. F1 hybrids of wild and domesticated rice species typically show better qualities for farming (hybrid vigor) compared to their parents; however, they are fully sterile. Thus, deletion of selfish toxins (like the one characterized in this manuscript) could ultimately result in better and fully fertile hybrid cultivars of this critical food staple.

The genetic dissection of this novel toxin-antidote element was performed to a very high standard, testing both for necessity and sufficiency of toxin and antidote functions. The combination of mutant alleles generated in the NIL background and the natural genetic variation found in *O. sativa* (premature stop codon in the toxin and transposon insertion in the promoter region of the antidote) convincingly shows that genes ORF3 and ORF5 code the underlying toxin and antidote, respectively. Overall, this is a very solid study. The manuscript is well written and the rationale behind all the experiments is well explained and justified. However, I have some major concerns, especially regarding the interpretation of this toxin-antidote element as an "ancient" element.

Major issues

1. Is ORF3/ORF5 truly an "ancient" TA element?

This manuscript claims that the ORF3/ORF5 is ancient. This claim is even included in the title of the manuscript. As far as I can tell, the evidence for this is very weak. In essence, the authors mention that:

Line 182:

"An extended protein sequence analysis to more distant species, including wheat and sorghum, suggests conservation of these two proteins, particularly the F-Box in ORF5 and DEXD and HELICc domains in ORF3 (Extended Data Figs. 17 and 18 and Supplementary Tables 7 and 8). Those results highlight that the ORF3-ORF5 pair may be an ancient TA mechanism that plays a role in maintaining species identity."

The fact that the toxin and the antidote have homology to genes found in other plant species is not enough evidence to claim that the TA is ancient. These could simply be "non-selfish" genes from which the TA evolved. Do the authors present any evidence, for example, that these two genes happen to be in linkage in any of these other plant species? If the genes are not in linkage, I find it hard to believe that they are acting as TAS. In the absence of any other evidence, I would refrain from including the word "ancient" in the title of this manuscript.

2. Background and references on toxin-antidote elements.

Toxin-antidote systems have been found before in many other species other than rice (beetles, nematodes, *saccharomyces pombe*, *neurospora*). However, the authors do not mention any of these studies neither in their introduction nor discussion. I think it would be important to contrast their findings to those in other systems (similarities and differences) and also to provide a better context to a wider audience.

Some examples of relevant papers that have not been cited include:

The first genetic description of a toxin-antidote element, which was identified in the beetle *Tribolium* (Beeman et al. Science 1992)

The first cloning of the underlying genes coding for the toxin and antidote (Seidel et al Science 2007; Seidel et al Plos Bio 2011)

The authors show in their manuscript that two rice TAs (qHMS1 and qHMS7) act independently in genetic crosses, causing increased male sterility. Analogous results have been previously described in nematodes, where TAs also act independently and their combined action causes extensive incompatibilities in both *C. elegans* (Ben-David et al, Science 2017) and *C. tropicalis* (Ben-David et al , Current Biology 2021).

It could also be worth mentioning, for instance, that more recently, TA's have been found in the model plant *Arabidopsis thaliana* (Simon et al. Genetics 2022).

3. Is there a reason why the authors refrain from giving specific names to the newly discovered toxin and antidote pair? The current names (ORF3 and ORF5) reflect the total number of (candidate) genes found in the introgressed region. However, I believe this will certainly generate confusion because the other TA locus also mapped by this group in rice (qqHMS7) is made up of a toxin-antidote pair named ORF3 and ORF2. Thus, unless I'm missing something, there are two evolutionary unrelated genes coding for toxins in rice that have exactly the same name: ORF3.

Minor issues:

1. The term "AA genome" as used in this manuscript is confusing. Although the authors do mention in line 47 of the introduction that "There are 9 *Oryza* species in the AA genome including...". At no point the authors define what AA stands for. Also, it would be much more accurate to describe it as the "AA genome group" and not "AA genome" because otherwise readers may think that it makes reference to an individual genome assembly.

2. There is no need to use the acronym "RI" for reproductive isolation (line 22). This term is only used a few times throughout the manuscript.

3. In line 201 of the discussion the authors write: "... suggesting that the TAs together can build up partial hybrid female sterility". Is this a typo? Shouldn't it be hybrid "male" sterility?

4. Line 101, the authors write "... confirming that it is the ORF3 from Mer that encodes an effective toxin". Perhaps "functional" is a better word choice than "effective"?

5. Line 116, mention what are those specific domains.

6.. Line 163 "... an early stop due to the G-T switch". Please use more accurate terminology, such as G to T mutation or even "transversion" instead of "switch". Same logic applies to line 170

7. 4. The authors write in their abstract:

"Unlike qHMS7 that originally had no function in Mer but gained a functional TA system during evolution, however, qHMS1 originally existed in Mer but evolved non-functional versions during

speciation.”

The phrasing “ qHMS1 originally existed in Mer but evolved non-functional versions during speciation” is a bit ambiguous. It sounds like qHMS1 was active in Mer populations in the past but is now inactive (in Mer). It may be more accurate to say that Mer carries a functional allele of qHMS1 whereas this locus is non-functional in *O. sativa* DJY1.

Reviewer #3:

Remarks to the Author:

This MS deals with the discovery of a second selfish genetic element conferring hybrid sterility in crosses between cultivated rice (DJY1-*Oryza sativa*) and wild *O. meridionalis* (Mer). This research and related paper largely recall (in methods and questions, and even in the format of the figures and data presentation) an extremely similar study by the same team (Yu XW, et al. A selfish genetic element confers non-Mendelian inheritance in rice. 2018, Science). This time, by using an identical approach, the authors find that at another locus, qHMS1, an ancient Mer allele kills the pollen carrying the newly evolved DJY1 allele. This is opposite to the previous finding where the newly evolved DJY1 allele at the qHMS7 locus kills the pollen carrying the ancient Mer allele. Results and experiments, as in the previous paper, are straightforward and provide extremely robust and compelling evidence. There is no evident flaw and I really have nothing to add from this side. While the MS evidently lacks innovation in question and methods and sounds incremental respect to previous work, still the findings are important as the new discovered toxin-antidote (TA) element qHMS1 has a different evolutionary history compared to the former one (i.e. qHMS7), being of very recent origin. This, coupled with its polymorphic stage, questions on its effective rule as speciation gene (and correctly, the authors do not put much emphasis on this aspect). However, to strength the impact of their new findings, in this paper I would expect the authors have comparatively discussed that these two selfish genetic elements have a very different (and independent) origins (i.e. involving two very different toxin proteins, a RIP and an helicase), the way they may have initially evolved (probable as killer meiotic drivers), the different location of respective antidotes (nuclear and mitochondrial), their different impact on reproductive isolation between species. In other words, I suggest the authors to tell this new chapter of their mission to clone all the major loci underlying HMS between Mer and cultivated rice from a different perspective than the one already told in their previous paper (and maybe in the next one). This seems essential to me for attracting the interest of readers and citations also on this second paper.

Reviewers' Comments:

Reviewer #1:

Remarks to the Author:

I found this version improved a lot. However, the revised version also introduces many new problems. I still have several questions to discuss with.

Major

1. The abstract is not clear enough and should be rewritten.

Line 22-23 "There are four major quantitative trait loci governing hybrid male sterility between cultivated rice (*Oryza sativa*) and *O. meridionalis* (Mer)". Where does the data come from? It would be very confused to see this statement without any background information in the abstract.

Line 25-26 "that encodes a toxin-antidote system distinct from the one encoded by qHMS7, another locus we cloned previously". This description is ambiguous because both qHMS1 and qHMS7 are toxin-antidote systems.

Line 27. The description of "none-functional qHMS1" and "functional qHMS1 and qHMS7" is confusing, especially for a locus with two genes.

Line 31. What does "distinct toxin-antidote systems" and "tiered reproductive isolation" mean?

In the abstract, I did not find any information about the "Evolutionary Breakdown" that mentioned in the title.

2. In the introduction, some descriptions were not accurate and were not adequately cited. For example, Line 37-39, differentiations in flower habit may also lead to pre-zygotic barriers. As reported, the color of monkeyflower and the corresponding pollination mode contribute to reproductive barriers. Line 45-49, the authors described that "The genetic mechanisms of HS often involve the toxin-antidote system". However, systems for HS are far more than the toxin-antidote system. They might be described as "responder", "distorter", "killer", and "protector", and all these definitions have particular meanings. Similar problems existed in the discussion part.

Line 55-56, "the rice S27/S28, DPL1/DPL2 and DGS1/DGS2 gene pairs kill pollens that carry none-functional antidote alleles". These cases have nothing to do with the antidote function.

Line 60-61. I read the linked reference and it seems that only the qHMS7 case was defined as the toxin-antidote system.

Line 62-63. These multiple-gene systems may not be simply classified as "none-functional locus".

3. Line 139-140. The knock-out mutant of ORF3 might be described and explained in detail because this is the target gene of qHMS1 locus. In addition, so does ORF5.

4. Line 218-224. This result should be expanded a lot. The related mechanism remains unclear.

5. May describe the frequency of different alleles in different species. In Figure 5a, the genotype frequency of the functional HPT1 and HPA1 is very low in the BB (1/3) and CC (1/5) rice species. In this case, the evidence of "loss-of-function of HPT1 is a recent evolutionary event" has not been strong enough. Whether HPT1 and HPA1 are ancient remains uncertain. Accordingly, "Evolutionary Breakdown" in the title is not convince enough.

6. As indicated by the reviewer, this is not the first toxin-antidote system in rice. Therefore, it is not proper to name the genes Hybrid Pollen Toxin 1 and Hybrid Pollen Antidote 1.

7. Figure S16. I think this result is still not persuasive.

Minor

Figure S7, there are different types of arrowheads in the figure and may lead to misunderstanding.

Figure 1g, means \pm SD?

Figure S9a, the name of the ORFs should be labeled upwards. Where is the citation of Figure S9b?

Figure S10. The title is not smooth.

The information of figure S12a is included in figure 1h. The figure S12c and figure 1h is also redundant.

Figure 4b. The color of the words in the construct is difficult to distinguish.

Table S2. The data rather than description should be presented in the table. What does "Expected ratio" and "Expected value" mean?

Table S3. What does two "Expected ratio" mean?

Tables S4, S5, S6. I am confused that why these tables presented separately?

Line 129, "615-bp deletion and five SNPs in ORF4", this 615-bp deletion is located in the intron and does not change the protein sequence?

Line 174, what does "S9-S11" mean? There is no explanation in figure S13a. The abbreviation should be explained on first occurrence.

Line 176, what is the developmental stages of the sample?

Line 213-216. The genotype and phenotype of the plants that used for RNA-sequencing should be described. Note that the specific allele of the genes should indicated clearly.

Line 222. Table S10 was cited before the other tables.

Reviewer #2:

Remarks to the Author:

After carefully reviewing the authors' response, I am pleased to note that the authors have successfully addressed my comments and those of the other two reviewers. . As a result, the manuscript now reads more smoothly, making it significantly easier to grasp its novelty and comprehend its impact in a wider context. While the molecular mechanism of qHMS1 still remains a mystery (the absence of a physical interaction between the toxin and antidote is puzzling, for instance), the genetic work conducted is remarkably thorough, well-executed, and holds significant value. I am very curious to know how exactly these two pollen killers could potentially lead to partial female gamete sterility (Fig. 6g). However, this is a very intriguing observation that warrants a follow up study!

With no further comments to make, I recommend this manuscript for publication

Reviewers' Comments:

Reviewer #1:

Remarks to the Author:

I found this version improved a lot. However, the revised version also introduces many new problems. I still have several questions to discuss with.

Major

1. The abstract is not clear enough and should be rewritten.

Line 22-23 "There are four major quantitative trait loci governing hybrid male sterility between cultivated rice (*Oryza sativa*) and *O. meridionalis* (Mer)". Where does the data come from? It would be very confused to see this statement without any background information in the abstract.

Line 25-26 "that encodes a toxin-antidote system distinct from the one encoded by qHMS7, another locus we cloned previously". This description is ambiguous because both qHMS1 and qHMS7 are toxin-antidote systems.

Line 27. The description of "none-functional qHMS1" and "functional qHMS1 and qHMS7" is confusing, especially for a locus with two genes.

Line 31. What does "distinct toxin-antidote systems" and "tiered reproductive isolation" mean?

In the abstract, I did not find any information about the "Evolutionary Breakdown" that mentioned in the title.

2. In the introduction, some descriptions were not accurate and were not adequately cited. For example, Line 37-39, differentiations in flower habit may also lead to pre-zygotic barriers. As reported, the color of monkeyflower and the corresponding pollination mode contribute to reproductive barriers. Line 45-49, the authors described that "The genetic mechanisms of HS often involve the toxin-antidote system". However, systems for HS are far more than the toxin-antidote system. They might be described as "responder", "distorter", "killer", and "protector", and all these definitions have particular meanings. Similar problems existed in the discussion part.

Line 55-56, "the rice S27/S28, DPL1/DPL2 and DGS1/DGS2 gene pairs kill pollens that carry none-functional antidote alleles". These cases have nothing to do with the antidote function.

Line 60-61. I read the linked reference and it seems that only the qHMS7 case was defined as the toxin-antidote system.

Line 62-63. These multiple-gene systems may not be simply classified as "none-functional locus".

3. Line 139-140. The knock-out mutant of ORF3 might be described and explained in detail because this is the target gene of qHMS1 locus. In addition, so does ORF5.

4. Line 218-224. This result should be expanded a lot. The related mechanism remains unclear.

5. May describe the frequency of different alleles in different species. In Figure 5a, the genotype frequency of the functional HPT1 and HPA1 is very low in the BB (1/3) and CC (1/5) rice species. In this case, the evidence of "loss-of-function of HPT1 is a recent evolutionary event" has not been strong enough. Whether HPT1 and HPA1 are ancient remains uncertain. Accordingly, "Evolutionary Breakdown" in the title is not convince enough.

6. As indicated by the reviewer, this is not the first toxin-antidote system in rice. Therefore, it is not proper to name the genes Hybrid Pollen Toxin 1 and Hybrid Pollen Antidote 1.

7. Figure S16. I think this result is still not persuasive.

Minor

Figure S7, there are different types of arrowheads in the figure and may lead to misunderstanding.

Figure 1g, means \pm SD?

Figure S9a, the name of the ORFs should be labeled upwards. Where is the citation of Figure S9b?

Figure S10. The title is not smooth.

The information of figure S12a is included in figure 1h. The figure S12c and figure 1h is also redundant.

Figure 4b. The color of the words in the construct is difficult to distinguish.

Table S2. The data rather than description should be presented in the table. What does "Expected ratio" and "Expected value" mean?

Table S3. What does two "Expected ratio" mean?

Tables S4, S5, S6. I am confused that why these tables presented separately?

Line 129, "615-bp deletion and five SNPs in ORF4", this 615-bp deletion is located in the intron and does not change the protein sequence?

Line 174, what does "S9-S11" mean? There is no explanation in figure S13a. The abbreviation should be explained on first occurrence.

Line 176, what is the developmental stages of the sample?

Line 213-216. The genotype and phenotype of the plants that used for RNA-sequencing should be described. Note that the specific allele of the genes should indicated clearly.

Line 222. Table S10 was cited before the other tables.

Reviewer #2:

Remarks to the Author:

After carefully reviewing the authors' response, I am pleased to note that the authors have successfully addressed my comments and those of the other two reviewers. . As a result, the manuscript now reads more smoothly, making it significantly easier to grasp its novelty and comprehend its impact in a wider context. While the molecular mechanism of qHMS1 still remains a mystery (the absence of a physical interaction between the toxin and antidote is puzzling, for instance), the genetic work conducted is remarkably thorough, well-executed, and holds significant value. I am very curious to know how exactly these two pollen killers could potentially lead to partial female gamete sterility (Fig. 6g). However, this is a very intriguing observation that warrants a follow up study!

With no further comments to make, I recommend this manuscript for publication

Response to Reviewer

Reviewer #1:

I found this version improved a lot. However, the revised version also introduces many new problems. I still have several questions to discuss with.

Response: We highly appreciate the Reviewer's time and efforts in reviewing our manuscript. The Reviewer's comments have helped a lot to improve our work. We have addressed all the comments in this revised version and hope that our revision is moving to a satisfactory degree.

Major

1. The abstract is not clear enough and should be rewritten.

Line 22-23 "There are four major quantitative trait loci governing hybrid male sterility between cultivated rice (*Oryza sativa*) and *O. meridionalis* (*Mer*)". Where does the data come from? It would be very confused to see this statement without any background information in the abstract.

Response: Thanks for the valuable point. The information on number of major quantitative trait loci is from previously published work, but the journal policy does not allow citation of references in Abstract. The related paper was cited in Introduction. To avoid confusion, we have changed the sentence to "Hybrid sterility, a major form of reproductive isolation exists between cultivated rice (*Oryza sativa*) and *O. meridionalis* (*Mer*), the far distant progenitor of cultivated rice in the AA genome group".

Line 25-26 "that encodes a toxin-antidote system distinct from the one encoded by *qHMS7*, another locus we cloned previously". This description is ambiguous because both *qHMS1* and *qHMS7* are toxin-antidote systems.

Response: Thanks for the valuable comment. We have changed the sentence to "Here we report isolation of *qHMS1*, a quantitative trait locus controlling hybrid male sterility between these two species. Like *qHMS7*, another locus we cloned previously, *qHMS1* also encodes a toxin-antidote system, but differs in the encoded proteins, their

evolutionary origin and action time point during pollen development”.

Line 27. The description of “none-functional *qHMS1*” and “functional *qHMS1* and *qHMS7*” is confusing, especially for a locus with two genes.

Response: Thanks for the valuable point. We have changed “none-functional *qHMS1*” to “*qHMS1*-D, an allele from cultivated rice”, and “functional *qHMS1* and *qHMS7*” to “*qHMS1*-*Mer* and *qHMS7*-D”.

Line 31. What does “distinct toxin-antidote systems” and “tiered reproductive isolation” mean?

Response: Thanks for the valuable point. Here “distinct toxin-antidote systems” means that these two toxin-antidote systems are different in the encoded proteins, their evolutionary origin and when they kill pollens after meiosis. “Tiered reproductive isolation” means that these two systems work in a stepwise manner, that is, *qHMS1* kills half of pollens at earlier developmental stage and *qHMS7* kills 50% pollens of the remaining viable half at later stage, such that finally 75% pollens in total are killed as a combined action. To make it more understandable, we have changed “Our results indicate that distinct toxin-antidote systems provide tiered reproductive isolation for maintaining species identity and shed light on breakdown of hybrid male sterility” to “Our results indicate that different toxin-antidote systems provide stacked reproductive isolation for maintaining species identity and shed light on breakdown of hybrid male sterility”.

In the abstract, I did not find any information about the “Evolutionary Breakdown” that mentioned in the title.

Response: Thanks for the comment. We have changed the title to “A toxin-antidote system contributes to interspecific reproductive isolation in rice”.

2. In the introduction, some descriptions were not accurate and were not adequately cited. For example, Line37-39, differentiations in flower habit may also lead to

pre-zygotic barriers. As reported, the color of monkeyflower and the corresponding pollination mode contribute to reproductive barriers.

Response: Thanks for the valuable comments. Based on your suggestions, we have now revised these parts as following: “In plants, the pre-zygotic barriers include geographical isolation, differentiations in flower habit, failure in cell-to-cell recognition between the pollen and stigma, inability in pollen germination and pollen tube growth, and defects in cell fusion between sexual gametes. For example, the speciation locus *YUP* in mokeyflowers¹, the unilaterally incompatibility system found in *Brassicaceae*^{2,3}, *Zea mays*⁴⁻⁷ and tomato⁸⁻¹⁰ is the well studied pre-zygotic mechanism”. (Lines 39-44)

Line 45-49, the authors described that “The genetic mechanisms of HS often involve the toxin-antidote system”. However, systems for HS are far more than the toxin-antidote system. They might be described as “responder”, “distorter”, “killer”, and “protector”, and all these definitions have particular meanings. Similar problems existed in the discussion part.

Response: Thanks for the comment. Following the suggestion, we have revised the text as “The toxin-antidote system (sometimes called killer-protector, meiotic drive, gene drive and segregation distortion in different literatures) is one of the genetic mechanisms conferring HS in animals and plants¹¹. Genetic elements belonging to this group include the *segregation distorter (SD)* system in *Drosophila*^{12,13}, the *wtf* loci in yeast^{14,15}, the *peel-1/zeel-1* and *sup35/pha-1* genes in nematodes¹⁶⁻¹⁸, the *t*-haplotype in mice¹⁹⁻²¹, the *Spok*²² and *het-s* genes²³ in *Podospora anserina*, the *Sk* genes in *Neurospora*²⁴, and *APOK3* in Arabidopsis²⁵”. (Lines 47-54)

Line 55-56, “the rice *S27/S28*, *DPL1/DPL2* and *DGS1/DGS2* gene pairs kill pollens that carry none-functional antidote alleles”. These cases have nothing to do with the antidote function.

Response: Thanks for the excellent point. We have revised the text as “In rice, the *S27/S28*²⁶, *DPL1/DPL2*²⁷ and *DGS1/DGS2*²⁸ are pairs of duplicated genes and pollens

carrying malfunctioned alleles at both of the paired loci are defective in development, leading to HS in hybrid plants. The genes encoding those systems are usually located in separated loci and how they were evolved during speciation has been hypothesized in the Bateson-Dobzhansky-Muller model^{29,30}". (Lines 60-64)

Line 60-61. I read the linked reference and it seems that only the *qHMS7* case was defined as the toxin-antidote system.

Response: Thanks for the comment. Right, among those references, only the term “toxin-antidote” system was used for *qHMS7*. For the *S1* and *S5* loci, the authors used the term “killer-protector” to describe the systems. Those HS-related loci (including *Sa* and *Sc*) are similar in mode-of-action, essentially belonging to the same group. The diversity of terms in describing those systems and toxin-antidote terminology have been clarified in a recent review (Toxin-antidote elements across the tree of life. *Annu. Rev. Genet.* 2020. 54:387–415). We have changed “toxin-antidote” to “toxin-antidote (or killer-protector)” for better readability. (Lines 65-66)

Line 62-63. These multiple-gene systems may not be simply classified as “none-functional locus”.

Response: Thanks for the comment. We have changed “none-functional locus” to “non-functional antidote or protector genes”. (Line 68)

3. Line 139-140. The knock-out mutant of *ORF3* might be described and explained in detail because this is the target gene of *qHMS1* locus. In addition, so does *ORF5*.

Response: Thanks for the comment. We have revised the text as “Then we focused on *ORF3* first and produced *ORF3* knock-out lines in the DJY1/NIL-*qHMS1* F₁ background using the CRISPR-Cas9 technology. We investigated three lines each representing a unique editing type in T₀ or T₁ plants and found that loss-of-function mutation of *ORF3* rescued pollen fertility (Fig. 2a-f). As expected, the segregation of three genotypes at *qHMS1* fits into the 1:2:1 ratio in the self-bred offspring of the knockout lines and the DJY1 allele of *qHMS1* was successfully transmitted to

progeny (Fig. 2g, Supplementary Table 2)”. (Lines 148-154)

For *ORF5*, we revised the text as “We also used the CRISPR-Cas9 technology to produce loss-of-function mutation of *ORF5* in the same DJY1/NIL-*qHMSI* F₁ background and found that the knock-out line showed complete sterility, instead of ~50% male sterility (Fig. 3b, e). Subsequently, we knocked out *ORF5* in both DJY1 (homozygous for the *ORF5-D* allele and fully fertile) and NIL-*qHMSI* (homozygous for the *ORF5-Mer* allele and fully fertile) and found that the DJY1^{*orf5*} plant was still fully fertile but the NIL-*qHMSI*^{*orf5*} plant lost pollen fertility completely (Fig. 3a-d)”. (Lines 167-173)

4. Line 218-224. This result should be expanded a lot. The related mechanism remains unclear.

Response: Thanks for the constructive suggestion. We performed a new RNA-seq experiment on the anthers at the uninucleate stage, followed by quantitative RT-PCR confirmation, and then conducted extensive analysis in combination with the existing RNA-seq data generated using anthers at the meiotic stage. Our analysis provides insight into mechanisms of how the toxin HPT interferes pollen development. We revised the text by adding the results from the new analysis as “In order to elucidate the relevant molecular mechanism of how *qHMSI* affects male gametogenesis, we performed RNA-seq experiments on anthers from the *gHPT-COM* transgenic plant carrying the *HPT-Mer* genetic fragment and displaying full pollen sterility in the DJY1 background, and DJY1 as a control (Supplementary Fig. 18a, b). The anthers were sampled at the meiotic stage (S8-S9) and the uninucleate stage (S9-S10) at which the first mitosis was arrested (Supplementary Fig. 6). The results showed that there were 3,520 differentially expressed genes (DEGs) at the meiotic stage, with 2,103 down-regulated and 1,417 up-regulated (Supplementary Fig. 18c). Among the 2,103 DEGs, 61 were down-regulated by 20 folds, of which 21 are anther-specific expressed (Supplementary Table 4; <http://rapdb.dna.affrc.go.jp>). At the uninucleate stage, there were 4,905 DEGs, with 2,717 down-regulated and 2,188 up-regulated (Supplementary Fig. 18d). Among the 2,717 DEGs, 20 were down-regulated by 20

folds, of which 7 are anther-specific expressed (Supplementary Table 4). Then we combined data from the two stages and identified 1,953 co-DEGs, with 1,243 down-regulated and 710 up-regulated (Supplementary Fig. 18e, f). GO analysis on those 1,243 genes indicated that they were clustered in various pathways and cellular processes, particularly in the processing and metabolism of ncRNA and rRNA (Supplementary Fig. 18g). KEGG analysis on the same DEGs revealed various metabolic pathways, especially in the sugar and starch metabolisms (Supplementary Fig. 18h, Supplementary Fig. 19). We performed qRT-PCR on some of those glycolysis or gluconeogenesis-related DEGs and confirmed their down-regulated expression in the *gHPT-COM* transgenic plant (Supplementary Fig. 20). Together, those results suggest a possible role of HPT in interfering expression of sugar/starch metabolic genes, which in turn may affect subsequent starch granule formation during pollen development, leading to final pollen abortion.

Supplementary Fig. 18. RNA-Seq analysis.

a, b, Pollen phenotype at maturity stage of DJY1 (a) and *gHPT-COM* transgenic plant

(b). Scale bar, 100 μm . *gHPT-COM* transgenic plants were obtained by transferring *HPT* of *Mer* into DJY1 callus. Anther samples used for RNA-seq analysis were collected at meiotic stage (S8-S9) and uninucleate stage (S9-S10). **c, d**, Scatter plots of differentially expressed genes (DEGs) at meiotic stage (S8-S9) (**c**) and uninucleate stage (S9-S10) (**d**) with the base 2 logarithm fold change and probability of p -value < 0.05 . The expression of genes in DJY1 was treated as control. The blue dots indicate the down-regulated genes, and the red dots indicate the up-regulated genes and the grey dots indicated the genes with no difference. **e, f**, Venn of Co-down-regulated (**e**) and co-up-regulated (**f**) genes analysis between meiotic stage and uninucleate stage. **g**, Enrichment of Gene Ontology (GO) terms of down-regulated differentially expressed genes (DEGs). **h**, Kyoto Encyclopedia of Genes and Genomes (KEGG) pathways of down-regulated DEGs.

Supplementary Fig. 19 Metabolic pathway of glycolysis/gluconeogenesis pathway. Metabolic pathway of glycolysis/gluconeogenesis pathway in the KEGG pathway database (<http://www.genome.jp/kegg/>). The down-regulated genes involved in glycolysis /gluconeogenesis pathway were marked in red.

b

RAP LOCUS	MSU LOCUS	Name	Description
Os01g0190400	LOC_Os01g09460	OsHXK8	Similar to Hexokinase
Os03g0381000	LOC_Os03g26430	RMP9	Similar to Aldose 1-epimerase-like protein
Os06g0256500	LOC_Os01g09461	Pgi2(Pgib)	Similar to Glucose-6-phosphate isomerase
Os02g0714200	LOC_Os01g09462	OsPFPA1	Similar to Pyrophosphate--fructose 6-phosphate 1- phosphotransferase alpha subunit (EC 2.7.1.90) (PFP)
Os06g0326400	LOC_Os01g09463	OsPFPA2	Phosphofructokinase domain containing protein
Os03g0330200	LOC_Os01g09464	-	2,3-bisphosphoglycerate-independent phosphoglycerate mutase (EC 5.4.2.1) (Phosphoglyceromutase)
Os12g0145700	LOC_Os01g09465	-	Pyruvate kinase family protein
Os06g0104900	LOC_Os01g09466	-	Similar to L-lactate dehydrogenase B (EC 1.1.1.27) (LDH-B) (Fragment)
Os07g0693100	LOC_Os01g09467	pdc3*	Similar to Pyruvate decarboxylase isozyme 3 (EC 4.1.1.1) (PDC)
Os02g0105200	LOC_Os01g09468	-	Similar to Dihydroliipoamide S-acetyltransferase (EC 2.3.1.12)

Supplementary Fig. 20 Genes expression verification.

a, qRT-PCR expression analysis of Genes involved in glycolysis/gluconeogenesis pathway. These genes were down-regulated in anther of *gHPA-COM* transgenic plants compared to DJY1. The anthers were analyzed at the meiotic stage (S8-S9) and the uninucleate stage (S9-S10). *UBQUITIN1* (*LOC_Os03g13170*) was used as an internal control. Data are means \pm SD ($n = 3$). * $p < 0.05$ and ** $p < 0.01$ in student's *t*-test. **b**, Description of these genes.

5. May describe the frequency of different alleles in different species. In Figure 5a, the genotype frequency of the functional HPT1 and HPA1 is very low in the BB (1/3) and CC (1/5) rice species. In this case, the evidence of “loss-of-function of HPT1 is a recent evolutionary event” has not been strong enough. Whether HPT1 and HPA1 are

ancient remains uncertain. Accordingly, “Evolutionary Breakdown” in the title is not convince enough.

Response: Thanks for the valuable point. Based on the limited number of accessions we examined, the frequency of *HPT* and *HPA* is low in the BB (1/3) and CC (1/5) genome groups. We have tried to expend testing of these two groups, but failed to get extra accessions/collections. On the other hand, for other accessions in BB (2/3) and CC (4/5) groups we were unable to amplify any DNA fragment from either *HPT* or *HPA* genomic region for unknown reason. For all accessions (wild and cultivated) from which *HPT* was successfully amplified, non-functional *HPT* was found only in cultivated rice and its immediate progenitor. In view of those considerations, we changed “loss-of-function of *HPT1* is a recent evolutionary event” to “loss-of-function of *HPT* might occur more recently during evolution of the AA genome”, and removed “Evolutionary Breakdown” from the title.

6. As indicated by the reviewer, this is not the first toxin-antidote system in rice. Therefore, it is not proper to name the genes Hybrid Pollen Toxin 1 and Hybrid Pollen Antidote 1.

Response: Thanks for the point. We have changed “*HATI*” and “*HPAI*” to “*HPT*” and “*HPA*”, respectively.

7. Figure S16. I think this result is still not persuasive.

Response: Thanks for the comment. We agree with that the *in situ* hybridization result is below a satisfactory degree. We have repeated the experiment again and presented the new result in Supplementary Fig. 16 (also copied below for Reviewer’s convenience). Although the results were improved slightly, the contrast between NIL-*qHMS1* and the controls is not persuasive enough. With this in mind, we looked up literatures reporting RNA *in situ* hybridization on rice anthers at uninucleate stage to learn where we could improve the protocol. We found one where the *in situ* hybridization image is also not striking in anthers at this stage, compared to clearer results at earlier stages (Plant Cell, 2021, 23: 2226-2246). Therefore, it might be

challenging to conduct the RNA *in situ* hybridization on highly vacuolated pollens.

Supplementary Figure 16. RNA *in situ* hybridization analysis of *ORF5/HPA*.

a-d, Anther section from DJY1 was hybridized with antisense (**a, b**) or sense (**c, d**) probe. **b**, Magnification of box in **a**. **d**, Magnification of box in **c**. **e-h**, Anther section from NIL-*qHMS1* was hybridized with antisense (**e, f**) or sense (**g, h**) probe. **f**, Magnification of box in **e**. **h**, Magnification of box in **g**. Anthers were sampled at the uninucleate stage. Scale bar, 50 μ m.

Minor

Figure S7, there are different types of arrowheads in the figure and may lead to misunderstanding.

Response: Thanks for the excellent point. We have re-arranged the arrow and arrowhead to denote aborted and normal pollens.

Figure 1g, means \pm SD?

Response: Thanks for the point. Yes, SDs are included in Figure 1g. We have revised the corresponding legend accordingly.

Figure S9a, the name of the ORFs should be labeled upwards. Where is the citation of

Figure S9b?

Response: Thanks for the point. We have moved the labels above the bar and cited Figure S9b in the revised manuscript. (Lines 139-142)

Figure S10. The title is not smooth.

Response: Thanks for the valuable point. We have changed the figure title to “*ORF1*, *ORF2* and *ORF4* do not contribute to hybrid male sterility”.

The information of figure S12a is included in figure 1h. The figure S12c and figure 1h is also redundant.

Response: Thanks for the comments. We have removed figure S12a and figure S12c, and re-organized Supplementary figure 12 accordingly.

Figure 4b. The color of the words in the construct is difficult to distinguish.

Response: Thanks for the point. We have changed the filled color to make the words more visible.

Table S2. The data rather than description should be presented in the table. What does “Expected ratio” and “Expected value” mean?

Table S3. What does two “Expected ratio” mean?

Response: Thanks for the comment. We apologize for the confusion. “Expected ratio” means the segregation ratio according to Mendel's Law of Segregation. We have changed “Expected value” to “Expected number”, which means the numbers from calculation based on the 1:2:1 ratio. We provide explanation under the table.

Tables S4, S5, S6. I am confused that why these tables presented separately?

Response: Thanks for the point. We have combined the three tables into one (Supplementary Table 5) and provide reference information under the table.

Line 129, “615-bp deletion and five SNPs in *ORF4*”, this 615-bp deletion is located in

the intron and does not change the protein sequence?

Response: Thanks for the point. Yes, this deletion is located in the first intron and does not change the protein sequence according to annotation at the public database (Rice Genome Annotation Project (uga.edu)).

Line 174, what does “S9-S11” mean? There is no explanation in figure S13a. The abbreviation should be explained on first occurrence.

Response: Thanks for the point. “S9-S11” are used to indicate certain development stages of rice pollen. We have added relevant information in the revised text and legend.

Line 176, what is the developmental stages of the sample?

Response: The samples for transcription analysis were collected from plants at the flowering stage. We have added this information in the revised manuscript.

Line 213-216. The genotype and phenotype of the plants that used for RNA-sequencing should be described. Note that the specific allele of the genes should indicated clearly.

Response: Thanks for the excellent comments. We have provided relevant information in the revised text. (Lines 231-236)

Line 222. Table S10 was cited before the other tables.

Response: Thanks for the point. We have corrected this citation.

Reviewer #2:

After carefully reviewing the authors' response, I am pleased to note that the authors have successfully addressed my comments and those of the other two reviewers. As a result, the manuscript now reads more smoothly, making it significantly easier to grasp its novelty and comprehend its impact in a wider context. While the molecular mechanism of *qHMSI* still remains a mystery (the absence of a physical interaction between the toxin and antidote is puzzling, for instance), the genetic work conducted is remarkably thorough, well-executed, and holds significant value. I am very curious to know how exactly these two pollen killers could potentially lead to partial female gamete sterility (Fig. 6g). However, this is a very intriguing observation that warrants a follow up study!

With no further comments to make, I recommend this manuscript for publication

Response: We deeply appreciate the Reviewer's recognition to our work. With the Reviewer's encouragement, we are interested in exploring partial female gamete sterility conferred by combination of the two loci in future.

Reviewers' Comments:

Reviewer #1:

Remarks to the Author:

Most of my concerns are addressed, and I am satisfied with this version.

One minor issue is confusing:

Line 65-67, any evidence supports the toxin or antidote role of Sc-i or Sc-j? I am still confused with this description. In my understanding, the Sc-j is an essential gene for development and is not a toxin or antidote.

Response to Reviewer

Reviewer #1:

Most of my concerns are addressed, and I am satisfied with this version.

One minor issue is confusing:

Line 65-67, any evidence supports the toxin or antidote role of *Sc-i* or *Sc-j*? I am still confused with this description. In my understanding, the *Sc-j* is an essential gene for development and is not a toxin or antidote.

Response: We thank the reviewer for the kind reminding and appreciate the reviewer's expertise in this field. We went back and carefully read the cited paper. In brief, the *japonica* allele *Sc-j* of the *Sc* locus encodes a DUF1618 domain protein that is essential for pollen development, whereas at the *indica* allele *Sc-i*, the DUF1618-containing region is tandemly duplicated two or three times. In F₁ hybrids (*Sc-j/Sc-i*), *Sc-i* is expressed whereas *Sc-j* is suppressed, such that the pollens carrying *Sc-i* are selectively transmitted. Since the two alleles have distinct promoters, the authors proposed that *Sc-i* may work on the *Sc-j* promoter directly or indirectly to inhibit *Sc-j* expression. Therefore, although the action mode of *Sc-j/Sc-i* looks like a toxin-antidote system, *Sc-j/Sc-i* are actually homologs, and they do not encode a pair of toxin and antidote proteins. To be more informative, we added two sentences to the text: "In a special case, the two alleles of the *Sc* locus encode homologs of DUF1618 domain protein that is essential for pollen development, but are different in copy number and promoter region. In F₁ hybrids (*Sc-j/Sc-i*), *Sc-i* (more copies) is expressed whereas *Sc-j* (single copy) is suppressed, such that the pollens carrying *Sc-i* are selectively transmitted." (Lines 69-73)

Reviewers' Comments:

Reviewer #1:

Remarks to the Author:

I think this version of description is correct and I have no more question.

Response to Reviewer

Reviewer #1:

I think this version of description is correct and I have no more question.

Response: We thank the Reviewer's time and efforts in reviewing our manuscript and deeply appreciate the Reviewer's recognition to our work.